# Covariance Volume Maximization for Embodied Latent Exploration in Deep Reinforcement Learning

**Yiming Wang** [1,2]  **Yiheng Zhang** [1]  **Kaiyan Zhao** [3]  **Xingjie Zuo** [4]  **Xingyu Liu** [3]  **Xuetao Li** [3]  **Furui Liu** [5]  **Bo An** [2]
**Leong Hou U** [1]

## Abstract

Efficient exploration remains a key challenge in deep reinforcement learning, especially for embodied agents operating in realistic environments with high-dimensional observations and complex dynamics. Recent latent exploration methods define bonuses in a learned latent space, but often struggle in these settings where (i) representations can be noisy or policy-dependent, and (ii) common strategies such as randomized latent objectives or fixed directional spanning are brittle and fail to improve global coverage. We propose Covariance Volume Maximization (CVM), a coverage-driven latent exploration framework with two key components. First, we learn a behavioral state encoder using a *policy-mixture* objective to reduce representation drift under rapidly changing exploration policies, yielding stable and behaviorally meaningful latent displacements. Second, CVM rewards each transition by its exact increase in the log-determinant of the covariance of recent latent displacements, explicitly expanding the explored region and prioritizing under-covered directions. This objective coincides with the classical *D-optimal* design criterion, providing an information-efficiency justification. Extensive experiments on embodied navigation and manipulation tasks demonstrate that CVM substantially improves exploration efficiency and robustness, and scales effectively to different environments.

## 1. Introduction

Efficient exploration is a long-standing challenge in deep reinforcement learning, and is especially acute for *embodied* agents in realistic environments. Navigation and manipulation require exploring under high-dimensional sensory inputs and hard dynamics constraints, making broad coverage far harder than in low-dimensional benchmarks. In such settings, naive strategies such as count-based (Bellemare et al., 2016a; Tang et al., 2017b; Ostrovski et al., 2017b) and prediction error-based (Burda et al., 2018; Pathak et al., 2017; Badia et al., 2020b) bonuses quickly lose effectiveness, motivating representation learning and latent exploration.

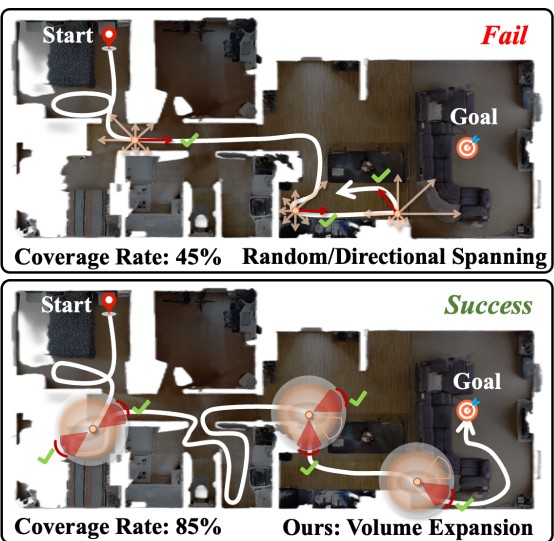

*Figure 1.* **Embodied navigation task: Top**: Random/directional spanning repeatedly probes directions that are infeasible under environment and dynamics constraints, causing local cycling and getting stuck, which limits exploration (**45% coverage**) and fails to reach the goal. **Bottom**: Our volume expansion objective prioritizes low-covered reachable directions, yielding broader exploration (**85% coverage**) and successfully reaching the goal.

Recent work studies latent-space exploration bonuses that aim to induce structured novelty. A common recipe conditions the policy on a latent variable that indexes exploratory objectives to elicit diverse behaviors (Tirinzoni et al., 2025;

---

[1]University of Macau [2]Nanyang Technological University [3]Wuhan University [4]Imperial College London [5]Zhejiang Lab. Correspondence to: Furui Liu <liufurui@zhejianglab.com>, Leong Hou U <ryanlhu@um.edu.mo>.

*Proceedings of the 43rd International Conference on Machine Learning*, Seoul, South Korea. PMLR 306, 2026. Copyright 2026 by the author(s).

| Method | Encoder Type | Exploration Strategy | Rep. Robust. | Glob. Cov. | D-Opt. |
|---|---|---|---|---|---|
| RLE (Mahankali et al., 2024) | Generic | Random Latent Goals: $\phi(s_t)^\top z$ | ✓ | ✗ | ✗ |
| RIDE (Raileanu & Rocktäschel, 2020) | Dynamic-based | Latent Displacement: $\Delta\phi_t$ | ✓ | ✗ | ✗ |
| EME (Wang et al., 2024) | Metric-based | Latent Displacement: $\Delta\phi_t$ | ✗ | ✗ | ✗ |
| METRA (Park et al., 2023) | Metric-based | Directional Spanning: $\Delta\phi_t^\top z$ | ✗ | ✗ | ✗ |
| HILP (Park et al., 2024) | Metric-based | Directional Spanning: $\Delta\phi_t^\top z$ | ✗ | ✗ | ✗ |
| CVM (ours) | Metric-based | Volume Expansion: $\log(1 + \Delta\phi_t^\top \Sigma_{t-1}^{-1} \Delta\phi_t)$ | ✓ | ✓ | ✓ |

*Table 1.* Comparison of latent exploration methods. In **Exploration Strategy**, $\phi(\cdot)$ is the learned encoder, $\Delta\phi_t = \phi(s_{t+1}) - \phi(s_t)$, and $z$ denotes a sampled latent direction/goal. For CVM, $\Sigma_{t-1}$ is the regularized covariance of recent $\Delta\phi$ and the bonus is $\log(1 + \Delta\phi_t^\top \Sigma_{t-1}^{-1} \Delta\phi_t)$. **Rep. Robust.** (representation robustness) marks encoders explicitly stabilized against high-dimensional nuisances and policy drift. **Glob. Cov.** marks bonuses derived from an explicit *global coverage* objective that allocates exploration across latent directions based on accumulated coverage, rather than only local novelty or progress toward sampled goals. **D-Opt.** marks objectives that maximize a log-determinant (D-optimal (Pukelsheim, 2006)) criterion.

Sikchi et al., 2025). Within this framework, exploration is typically driven by (i) sampling random latent goals (Mahankali et al., 2024), (ii) rewarding the magnitude of successive latent displacements (Wang et al., 2024; Raileanu & Rocktäschel, 2020), or (iii) sampling a latent direction $z$ and rewarding progress along $z$ to span the latent space (Park et al., 2023; 2024). However, in realistic high-dimensional settings these approaches often degrade because the latent space is unstable: generic or random encoders lack control-relevant structure, dynamics-based encoders can overfit to nuisance appearance variation, and policy-dependent metric encoders drift as the exploration policy changes, making $\Delta\phi$ noisy and bonuses unreliable. More importantly, the resulting objectives still do not measure *global coverage*. Random-goal and displacement bonuses are local and direction-agnostic, so agents can cycle among nearby states without expanding the reachable set. Directional spanning partially addresses this, but is brittle under embodiment constraints: many sampled directions are redundant or physically infeasible under dynamics and contact, and optimization often collapses onto a few easy directions, leading to "stuck" behaviors and poor coverage in practice (Fig. 1). What is missing is a history-aggregated objective that explicitly expands under-covered latent dimensions while down-weighting redundant motion along already-covered ones (Fig. 1, bottom), motivating our volume-based criterion:

*This motivates shifting from directional to volume-based exploration: instead of rewarding progress along a sampled direction, we reward transitions that expand the volume of latent experience, directly promoting history-adaptive, embodiment-consistent global coverage.*

To address these limitations, we propose *Covariance Volume Maximization* (CVM) with two components. First, we learn a behavioral encoder that suppresses nuisance visual variation and preserves control-relevant dynamics; to prevent representation drift under rapidly changing exploration policies, we train it with a policy-mixture behavioral objective, yielding stable and consistent displacements $\Delta\phi$. Second,

CVM defines a coverage-driven exploration bonus: instead of rewarding local novelty or progress along sampled directions, it rewards each transition by its exact increase in the log-volume of the covariance ellipsoid of recent $\Delta\phi$, prioritizing under-explored directions and down-weighting redundant ones. Finally, cumulative CVM reward equals $\log \det(\Sigma_T)$, aligning with the *D-optimal* criterion in optimal experimental design (Pukelsheim, 2006) and providing an information-efficiency interpretation under linear-Gaussian assumptions.

Our main contributions are: (1) We propose CVM, a coverage-driven latent exploration bonus that rewards transitions by their incremental expansion of the covariance volume of recent latent displacements, emphasizing under-covered directions while down-weighting redundant progress. (2) We introduce a policy-mixture behavioral state encoder for realistic high-dimensional observations, mitigating representation drift under rapidly changing exploration policies and producing stable, behaviorally grounded latent displacements. (3) We prove that CVM optimizes a D-optimal log-determinant objective and connect it to optimal experimental design and information gain under standard assumptions. (4) We demonstrate consistent gains in exploration efficiency and robustness on embodied navigation and manipulation benchmarks.

## 2. Preliminaries

**Reinforcement Learning.** We model the environment as an MDP $\mathcal{M} = (\mathcal{S}, \mathcal{A}, P, r, \gamma)$, where $P(s'|s, a)$ denotes the transition dynamics, $r(s, a)$ the reward, and $\gamma \in [0, 1)$ the discount factor. A policy $\pi(a|s)$ induces trajectories by sampling $a_t \sim \pi(\cdot|s_t)$ and $s_{t+1} \sim P(\cdot|s_t, a_t)$. The objective is to learn $\pi$ that maximizes the expected discounted return $\mathbb{E}_\pi[\sum_{t=0}^\infty \gamma^t r(s_t, a_t)]$, typically via a value function $V^\pi(s) = \mathbb{E}_\pi[\sum_{t=0}^\infty \gamma^t r(s_t, a_t) \mid s_0 = s]$.

**Behavioral Metric-based Representation Learning.** Be-

**Inadequate exploration** ⟶ **Comprehensive coverage**

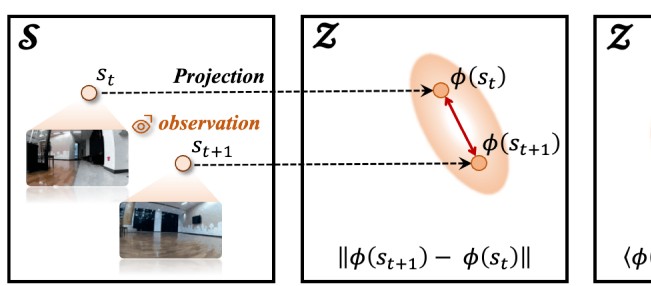
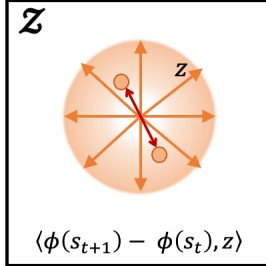
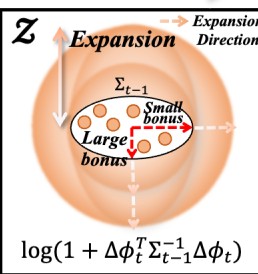

| Embodied Navigation | Latent displacement | Directional spanning | Volume expansion |

*Figure 2.* **CVM illustration and comparison to prior latent exploration bonuses.** A transition in the environment $s_t \to s_{t+1}$ is mapped by a state encoder $\phi$ into a latent displacement $\Delta\phi_t = \phi(s_{t+1}) - \phi(s_t)$ (left). Existing latent exploration bonuses typically either reward *direction-agnostic* latent change, e.g., the displacement magnitude $\|\Delta\phi_t\|$ (second panel), or reward progress along a *sampled* latent direction $z$, e.g., $\langle\Delta\phi_t, z\rangle$ (third panel). In contrast, CVM maintains the covariance $\Sigma_{t-1}$ of recent latent displacements and rewards a transition by its exact log-volume gain, $b_t = \log\left(1 + \Delta\phi_t^\top \Sigma_{t-1}^{-1} \Delta\phi_t\right)$ (right). This induces a coverage-seeking behavior: displacements aligned with already explored directions yield smaller bonuses, while steps that open under-covered directions enlarge the covariance ellipsoid and receive larger bonuses, promoting more comprehensive latent coverage.

havioral metric learning seeks an embedding where distances reflect *behavioral* similarity. Let $\phi_\omega : \mathcal{S} \to \mathbb{R}^n$ be a state encoder and define the latent distance $d_\phi(s_i, s_j) = \|\phi_\omega(s_i) - \phi_\omega(s_j)\|_2$. A behavioral metric specifies a target distance $d^\pi(s_i, s_j)$ that captures how differently two states evolve and are rewarded under policy $\pi$. Bisimulation-style metrics (Zhang et al., 2020; Castro et al., 2021) instantiate this idea by combining reward discrepancies with distances between next-state distributions. For example, MICo (Castro et al., 2021) defines a Bellman-style operator:

**Definition 1** (MICo Behavioral Metric Operator). *Let $\mathbb{M}$ be the space of bounded nonnegative functions $d : \mathcal{S} \times \mathcal{S} \to \mathbb{R}_{\geq 0}$. Define the MICo operator $\mathcal{F}_M^\pi : \mathbb{M} \to \mathbb{M}$ as*

$$\mathcal{F}_M^\pi(d)(s_i, s_j) = \mathbb{E}_{a_{i,j}\sim\pi}|r_i - r_j| + \gamma\mathbb{E}_{s'_{i,j}\sim P}d(s'_i, s'_j) \quad (1)$$

In practice, behavioral encoders are trained by minimizing the Bellman residual induced by this operator:

$$
\begin{aligned}
\mathcal{L}(\phi) = \mathbb{E}_{s_i, s_j \sim \mathcal{D}, s'_i \sim P_{s_i}^\pi, s'_j \sim P_{s_j}^\pi} \big[ & d_\phi(s_i, s_j) \\
& - |r_i - r_j| - \gamma d_\phi(s_{i+1}, s_{j+1}) \big]
\end{aligned}
\quad (2)
$$

where $\mathcal{D}$ is a replay buffer.

**Latent Exploration Bonus.** Latent exploration methods shape exploration in a learned representation by assigning an intrinsic bonus to states or transitions. Given an encoder $\phi : \mathcal{S} \to \mathbb{R}^n$, a typical bonus takes the form

$$b(s, s') = f\big(\phi(s), \phi(s')\big) \quad (3)$$

As summarized in Table 1, prior work largely falls into two families. *Random* latent exploration uses direction-agnostic objectives, e.g., sampling latent goals or rewarding the magnitude of consecutive latent displacements (Mahankali et al.,

2024; Raileanu & Rocktäschel, 2020; Wang et al., 2024); these encourage local novelty but do not explicitly control expansion across latent dimensions. *Directional spanning* instead samples a latent direction and rewards progress along it (Park et al., 2023; 2024), inducing diverse behaviors but coupling the signal to sampled directions rather than global coverage. This motivates replacing random- or direction-driven bonuses with a *global coverage-driven* objective; next we introduce CVM, which yields a simple, information-grounded exploration bonus.

## 3. Methodology

We present CVM, which combines a behavioral state encoder with a coverage-driven exploration bonus. We first learn a behavior-relevant latent space from high-dimensional observations, then define a CVM bonus that rewards transitions expanding the covariance volume of recent latent displacements. Finally, we connect the cumulative CVM objective to D-optimal design and information gain.

### 3.1. Behavioral State Encoder

**Motivation.** To handle high-dimensional observations in realistic environments, we first learn a state encoder $\phi_\omega : \mathcal{S} \to \mathbb{R}^n$ that maps observations to a compact latent space, reducing the effective space in which exploration is shaped. Prior latent exploration methods often rely on metric-based representation learning, such as Hilbert embeddings (Park et al., 2024), temporal-distance objectives (Park et al., 2023), or bisimulation-inspired metrics (Wang et al., 2024). However, in embodied exploration with complex visual inputs, these encoders can be sensitive to nuisance

variation and may yield latent displacements that do not reliably correspond to *behavioral* change. Our desideratum is therefore a representation that filters irrelevant visual noise and organizes the latent space by control-relevant structure, so that latent displacements capture **meaningful behavioral differences** of an embodied agent. Motivated by behavioral representation learning in RL (Zhang et al., 2020; Castro et al., 2021; Chen & Pan, 2022), we train $\phi_\omega$ with a behavioral-metric objective so that latent geometry reflects *behavioral* similarity: states are close when they induce similar rewards and controllable transitions, and far otherwise. This design provides two key benefits: **(i)** it suppresses nuisance visual variation by prioritizing control-relevant features over raw appearance, and **(ii)** it makes latent transitions $\Delta\phi = \phi(s') - \phi(s)$ more stable and behaviorally meaningful, which is crucial for constructing the CVM exploration bonus in the next section.

**Representation Drift under Exploration.** Previous bisimulation and RAP-style behavioral metrics-based latent exploration methods (Wang et al., 2024; Chen & Pan, 2022) are defined on-policy or strongly tied to a single policy $\pi$. When operating as an exploration-driven reward shaping mechanism, under which the policy changes rapidly during training. If the behavioral metric (and thus the encoder) is tightly coupled to the current policy, the learned latent space becomes a moving target: *as the policy evolves, the encoder geometry drifts accordingly*:

**Proposition 1** (On-policy objectives induce representation drift). *Let $\{\pi_t\}_{t \geq 0}$ be the evolving exploration policy. Suppose the encoder $\phi_\omega$ is trained to minimize an on-policy behavioral-metric loss*

$$\mathcal{L}_t(\omega) = \mathbb{E}_{(s,\tilde{s}) \sim \mu_{\pi_t}} \left[ \ell \big( d_\omega(s,\tilde{s}),\, d^{\pi_t}(s,\tilde{s}) \big) \right] \quad (4)$$

*where $\mu_{\pi_t}$ is the state-pair sampling distribution induced by $\pi_t$, $d_\omega(s,\tilde{s}) = \|\phi_\omega(s) - \phi_\omega(\tilde{s})\|_2$, and $d^{\pi_t}$ denotes a policy-conditioned behavioral metric. If $\pi_t \neq \pi_{t-1}$ and either $d^{\pi_t}$ or $\mu_{\pi_t}$ depends on $\pi_t$, then the minimizer set $\arg\min_\omega \mathcal{L}_t(\omega)$ generally differs from $\arg\min_\omega \mathcal{L}_{t-1}(\omega)$. Consequently, the induced latent geometry can drift over time even when the environment dynamics are fixed.*

**Corollary 1** (Representation drift corrupts latent displacements). *Define the latent displacement at time $t$ using the current encoder as $\Delta\phi_t = \phi_{\omega_t}(s_{t+1}) - \phi_{\omega_t}(s_t)$. If the encoder changes from $\omega_{t-1}$ to $\omega_t$, then for the same transition $(s_t, s_{t+1})$,*

$$\Delta\phi_t = \underbrace{\phi_{\omega_{t-1}}(s_{t+1}) - \phi_{\omega_{t-1}}(s_t)}_{\text{transition-induced displacement}}$$
$$+ \underbrace{\Big(\phi_{\omega_t}(s_{t+1}) - \phi_{\omega_{t-1}}(s_{t+1})\Big) - \Big(\phi_{\omega_t}(s_t) - \phi_{\omega_{t-1}}(s_t)\Big)}_{\text{representation drift}}$$
$$(5)$$

*The drift term can be non-negligible under rapidly changing exploration policies, causing $\Delta\phi_t$ to reflect representation updates rather than genuine behavioral change.*

Proposition 1 formalizes that on-policy behavioral objectives create a moving target as the policy evolves. Corollary 1 shows that this drift directly contaminates latent displacements $\Delta\phi_t$, this instability is particularly problematic for CVM, which estimates global coverage from recent latent displacements $\Delta\phi$. This motivates learning a representation-robust behavioral encoder before defining exploration bonuses on $\Delta\phi_t$.

**Mitigating drift via policy mixtures.** Proposition 1 and Corollary 1 suggest that the root cause of representation drift is the *policy dependence* of the metric target and the sampling distribution. We therefore replace the instantaneous policy $\pi_t$ with a *policy mixture* $\bar{\pi}_t$ that aggregates a set of recently visited behaviors. Concretely,

$$\bar{\pi}_t(a \mid s) = \sum_{k=1}^{K} w_k \, \pi_{t-k}(a \mid s) \quad (6)$$

where $\{\pi_{t-k}\}$ are recent policy snapshots (or, equivalently, an implicit mixture induced by the replay buffer), and $w_k \geq 0$, $\sum_{k=1}^{K} w_k = 1$, $\{w_k\}$ are fixed weights. Given $\bar{\pi}$, we define a bisimulation-style distance $d^{\bar{\pi}}$ as the fixed point of the Bellman operator:

$$\mathcal{F}^{\bar{\pi}}(d)(s,\tilde{s}) = \mathbb{E}_{\substack{a \sim \bar{\pi}(\cdot|s) \\ \tilde{a} \sim \bar{\pi}(\cdot|\tilde{s})}} [|r(s,a) - r(\tilde{s},\tilde{a})| \\ + \gamma\, \mathbb{E}_{\substack{s' \sim P(\cdot|s,a) \\ \tilde{s}' \sim P(\cdot|\tilde{s},\tilde{a})}} d(s',\tilde{s}')] \quad (7)$$

where $P$ is the transition kernel and $\gamma \in (0,1)$ is the discount factor. We then define the behavioral metric target under $\bar{\pi}_t$, i.e., $d^{\bar{\pi}_t}$ as the fixed point of the Bellman-style operator in (7), and train the encoder to match this mixture-policy metric.

**Proposition 2** (Policy mixtures reduce objective variation). *Assume the policy-conditioned metric $d^\pi$ is Lipschitz in $\pi$ with respect to a divergence $D(\cdot,\cdot)$, i.e., $\|d^\pi - d^{\pi'}\|_\infty \leq L\, D(\pi,\pi')$ for some $L > 0$.[1] Then the mixture metric varies more smoothly than the instantaneous metric:*

$$\left\| d^{\bar{\pi}_t} - d^{\bar{\pi}_{t-1}} \right\|_\infty \leq L \sum_{k=1}^{K} w_k\, D(\pi_{t-k}, \pi_{t-1-k}) \quad (8)$$

*In particular, if the policy updates are bounded on average, the right-hand side is smaller than the variation induced by using $\pi_t$ directly, yielding a more stable target for representation learning.*

---

[1]This assumption is standard when $d^\pi$ is defined as the fixed point of a contraction whose expectation is taken under $\pi$.

Proposition 2 formalizes the intuition that averaging over a set of recent behaviors attenuates rapid policy changes, turning an on-policy moving target into a slowly varying one. In practice, sampling transitions from a replay buffer implements this mixture implicitly: the encoder is trained on state-action pairs generated by multiple past policies, which improves learning stability and makes latent displacements $\Delta\phi_t$ less contaminated by representation drift.

### 3.2. Exploration via Covariance Volume Maximization

**Motivation.** Prior latent exploration bonuses mainly follow two strategies (Table 1). *Random* schemes reward sampled latent goals or latent displacement magnitude, which promotes local novelty but provides no mechanism to balance coverage across latent dimensions. *Directional spanning* instead samples a direction $z$ and rewards progress $\Delta\phi_t^\top z$, but its signal depends on the sampled directions and can waste exploration on already covered or infeasible directions under embodied constraints. These limitations motivate a direction-free objective that explicitly targets global coverage: As Fig. 2 shows, CVM rewards latent transitions that most expand the covariance volume of recent latent displacements, encouraging new directions while naturally down-weighting repeated ones.

**CVM Bonus.** Let $\phi_\omega : \mathcal{S} \to \mathbb{R}^n$ be the behavioral encoder from Section 3.1 and define the latent displacement

$$\Delta\phi_t = \phi_\omega(s_{t+1}) - \phi_\omega(s_t) \qquad (9)$$

CVM measures exploration progress by how much a transition expands the *volume* of the ellipsoid induced by the covariance of recent latent displacements. To this end, we maintain a covariance estimate $\Sigma_t \in \mathbb{R}^{n \times n}$ over recent $\Delta\phi$. Since CVM requires computing $\Sigma^{-1}$ (equivalently, $\log\det(\Sigma)$), $\Sigma_t$ must remain well-conditioned throughout training: early exploration or highly correlated displacements can make the empirical covariance nearly singular, leading to unstable inverses and spurious, high-magnitude bonuses. We therefore use an exponential moving average (EMA) update with step size $\beta$ to smooth the covariance estimates, and add a small ridge term $\lambda I$ to ensure $\Sigma_t$ is strictly positive definite:

$$\Sigma_t = (1-\beta)\Sigma_{t-1} + \beta\,\Delta\phi_t\Delta\phi_t^\top + \lambda I \qquad (10)$$

where $\beta \in (0,1]$ and $\lambda > 0$. At time $t$, before incorporating $\Delta\phi_t$ into the covariance, we define the CVM exploration bonus as the one-step log-volume gain induced by $\Delta\phi_t$,

$$b_t = \log\det(\Sigma_{t-1} + \Delta\phi_t\Delta\phi_t^\top) - \log\det(\Sigma_{t-1}) \quad (11)$$

By the matrix determinant lemma (Horn & Johnson, 2012), Eq. (11) admits an exact closed form:

$$b_t = \log(1 + \Delta\phi_t^\top \Sigma_{t-1}^{-1} \Delta\phi_t) \qquad (12)$$

**Lemma 1** (Mahalanobis interpretation). *Let $\Sigma \succ 0$ and $u \in \mathbb{R}^n$. The quadratic form $u^\top\Sigma^{-1}u$ equals the squared Mahalanobis distance (McLachlan, 1999) of $u$ under covariance $\Sigma$. Moreover, if $\Sigma = Q\Lambda Q^\top$ with $\Lambda = \mathrm{diag}(\lambda_1,\ldots,\lambda_n)$ and $\alpha \triangleq Q^\top u$, then*

$$u^\top\Sigma^{-1}u = \sum_{i=1}^{n} \frac{\alpha_i^2}{\lambda_i} \qquad (13)$$

*Consequently, components along low-variance directions (small $\lambda_i$) receive larger weight, while motion along high-variance directions is down-weighted.*

Lemma 1 makes the CVM mechanism explicit. Setting $u = \Delta\phi_t$ and $\Sigma = \Sigma_{t-1}$, the score $\Delta\phi_t^\top\Sigma_{t-1}^{-1}\Delta\phi_t$ is large when the current displacement opens directions that have small variance under past exploration (i.e., under-explored directions), and small when it repeats already expanded directions. Through the $\log(1 + \cdot)$ transform in Eq. (12), CVM further exhibits diminishing returns along directions that are already well covered.

### 3.3. Theoretical Analysis

We summarize key properties of the CVM bonus in Eq. (12). We analyze the idealized rank-one update $\Sigma_t = \Sigma_{t-1} + \Delta\phi_t\Delta\phi_t^\top$, while the EMA estimator in Eq. (10) serves as a smoothed and numerically stable implementation of the same principle.

**Proposition 3** (Global objective of CVM). *Assume $\Sigma_0 \succ 0$ and the rank-one update $\Sigma_t = \Sigma_{t-1} + \Delta\phi_t\Delta\phi_t^\top$. Define the per-step CVM bonus as the log-determinant increment (11), where the closed form follows from the matrix determinant lemma (Horn & Johnson, 2012). Then the cumulative bonus telescopes:*

$$\sum_{t=1}^{T} b_t = \log\det(\Sigma_T) - \log\det(\Sigma_0) \qquad (14)$$

*Consequently, maximizing cumulative CVM reward over a horizon $T$ is equivalent to maximizing the single global objective $\log\det(\Sigma_T)$.*

Proposition 3 clarifies the role of CVM: it is not a heuristic novelty score, but the exact marginal gain of a well-defined global coverage objective. This connection enables the geometric and information-theoretic interpretations below.

**Lemma 2** (Log-determinant and ellipsoid volume (Boyd & Vandenberghe, 2004)). *Let $\mathrm{Vol}(\cdot)$ denote the $n$-dimensional Lebesgue measure (Euclidean volume) on $\mathbb{R}^n$. For any $\Sigma \succ 0$, define the covariance ellipsoid $\mathcal{E}(\Sigma) \triangleq \{x \in \mathbb{R}^n : x^\top\Sigma^{-1}x \le 1\}$ and let $\mathbb{B}_n \triangleq \{x \in \mathbb{R}^n : \|x\|_2 \le 1\}$ be the unit Euclidean ball, then $\mathrm{Vol}(\mathcal{E}(\Sigma)) = \mathrm{Vol}(\mathbb{B}_n)\sqrt{\det(\Sigma)}$. Consequently,*

$$\log\mathrm{Vol}(\mathcal{E}(\Sigma)) = \log\mathrm{Vol}(\mathbb{B}_n) + \tfrac{1}{2}\log\det(\Sigma) \qquad (15)$$

*so* $\log \det(\Sigma)$ *is proportional to the log-volume of* $\mathcal{E}(\Sigma)$ *up to an additive constant.*

Lemma 2 gives $\log \det(\Sigma_T)$ a direct geometric meaning: maximizing it expands the volume of the explored displacement ellipsoid, encouraging balanced global coverage across directions. Moreover, Eq. (12) implies a saturation effect: as exploration accumulates, the displacement covariance $\Sigma$ grows, so displacements aligned with previously explored directions yield smaller gains, whereas displacements in under-covered directions remain strongly rewarded. Additional structural properties of the log-determinant objective (e.g., diminishing marginal gains under a set-selection view) are discussed in Appendix A.

**Proposition 4** (D-optimal design). *Let* $\Delta\phi_t \in \mathbb{R}^n$ *denote the latent displacement at transition* $t$, *and consider an auxiliary scalar response model for an unknown parameter* $\theta \in \mathbb{R}^n$:

$$y_t = \Delta\phi_t^\top \theta + \varepsilon_t \qquad (16)$$

*where* $\varepsilon_t \sim \mathcal{N}(0, \sigma^2)$. *Then the Fisher information matrix after* $T$ *transitions is*

$$\mathcal{I}_T(\theta) = \frac{1}{\sigma^2} \sum_{t=1}^{T} \Delta\phi_t \Delta\phi_t^\top \qquad (17)$$

*Consequently, maximizing* $\log \det(\Sigma_T)$ *is exactly the classical D-optimal design criterion for the estimation problem.*

Proposition 4 connects CVM to a classical notion of *information efficiency*. In optimal experimental design, D-optimality selects measurements so that the resulting parameter uncertainty shrinks as fast as possible in *all* directions, which is captured by maximizing the determinant of the information matrix (or equivalently minimizing the volume of the associated confidence ellipsoid). In our setting, $\Sigma_T$ plays an analogous role as an aggregated second-moment matrix of explored latent displacements. Thus maximizing $\log \det(\Sigma_T)$ encourages exploration that is informative across directions, rather than concentrating experience along a few redundant modes. This provides a principled justification for CVM: it allocates exploration effort to directions that most improve the agent's global knowledge of what displacements (and hence reachable states) are feasible under the environment dynamics.

**Summary.** CVM optimizes a single global objective, $\log \det(\Sigma_T)$, via exact per-step log-determinant gains, which expands the volume of explored latent displacements, yielding global, direction-balanced coverage with diminishing gains for redundant directions. Under standard linear-Gaussian assumptions, it aligns with D-optimal design (Pukelsheim, 2006), providing an information-efficiency interpretation for why CVM prioritizes under-explored directions that improve coverage the most. We outline the full training procedure in Algorithm 1.

# 4. Experiments

To evaluate the performance of CVM, we conduct experiments across a set of realistic embodied environments, assessing exploration efficiency, robustness, and scalability[2].

**Baselines.** We compare CVM against exploration baselines spanning bonus-based and latent exploration approaches. Specifically, we include: **(1) RND** (Burda et al., 2018), a standard prediction-error exploration bonus; **(2) RLE** (Mahankali et al., 2024), which encourages exploration via randomized latent objectives; **(3) RIDE** (Raileanu & Rocktäschel, 2020), which uses latent displacement learned from dynamics models as an exploration signal; **(4) EME** (Wang et al., 2024), which defines exploration bonuses from latent discrepancies in a bisimulation-style metric space; **(5) E3B** (Henaff et al., 2022), a count-based episodic bonus computed in an inverse-dynamics latent space; and two directional spanning methods, **(6) METRA** (Park et al., 2023) and **(7) HILP** (Park et al., 2024), which sample latent directions and reward progress along them to induce diverse behaviors.

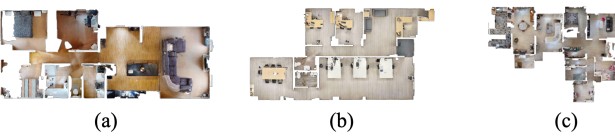

(a)        (b)        (c)

*Figure 3.* Three indoor environments with different settings (different number of floors and rooms) with different complexity level (easy, medium, hard) for the navigation task: (a) 1F6R-easy; (b)1F9R-medium; (c) 2F18R-hard.

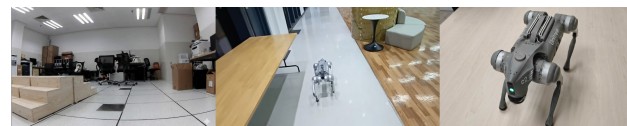

*Figure 4.* Navigation tasks evaluated in realistic indoor environments using the Unitree Go2 quadruped robot.

## 4.1. Experimental Setting

**Embodied Navigation Tasks.** We evaluate CVM on embodied navigation in both photorealistic simulation and real-world indoor environments. In simulation, we use Habitat (Savva et al., 2019) with the Habitat-Matterport 3D (HM3D) dataset (Yadav et al., 2023), where the agent must navigate from a randomized start to a goal in multi-room indoor scenes. As shown in Fig. 3, we select three environments with increasing layout complexity (Easy/Medium/Hard) (e.g., "1F6R-easy" indicates one floor and six rooms). For real-world evaluation, we de-

---

[2]The code is available at https://github.com/YimingWangMingle/CVM

| Environment | Nav-easy | Nav-medium | Nav-hard | Grasp | Handover | Targeted Handover |
|---|---|---|---|---|---|---|
| RND | 32.0%±7.5% | 25.2%±6.8% | 12.8%±5.6% | 47.1%±8.5% | 36.5%±6.8% | 19.1%±7.3% |
| RLE | 48.6%±6.8% | 35.2%±7.5% | 29.8%±5.9% | 57.1%±6.5% | 48.5%±6.8% | 32.5%±7.2% |
| RIDE | 41.3%±6.8% | 38.2%±7.5% | 25.8%±5.9% | 55.9%±9.5% | 44.8%±7.2% | 28.5%±8.3% |
| EME | 71.8%±11.2% | 68.6%±10.5% | 45.5%±12.1% | **82.2%**±**4.9%** | 69.4%±8.1% | 51.4%±7.8% |
| E3B | 75.8%±8.9% | 65.6%±9.5% | 37.5%±11.5% | 78.9%±7.5% | 58.8%±9.1% | 30.1%±6.5% |
| METRA | 55.8%±8.1% | 41.2%±9.1% | 29.8%±6.5% | 79.1%±7.2% | 63.1%±8.5% | 51.2%±5.8% |
| HILP | 61.2%±10.2% | 49.8%±9.1% | 30.5%±8.2% | 72.4%±8.2% | 65.2%±5.8% | 55.1%±6.7% |
| CVM | **78.7%**±**11.8%** | **71.9%**±**12.1%** | **69.2%**±**8.5%** | 81.2%±7.6% | **77.8%**±**6.9%** | **70.9%**±**7.5%** |

*Table 2.* Mean success rates comparison (averaged over 10 random seeds) in the of the embodied navigation and manipulation tasks.

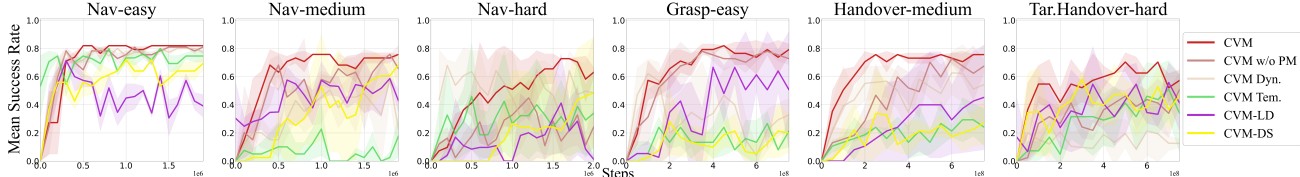

*Figure 5.* Mean success rate (10 seeds) comparison on CVM ablations. We compare CVM to encoder ablations and bonus replacements. Legend: **w/o PM** = without policy mixture; **Dyn.** = dynamics-based encoder; **Tem.** = temporal metric encoder; **LD** = latent displacement bonus; **DS** = directional spanning bonus.

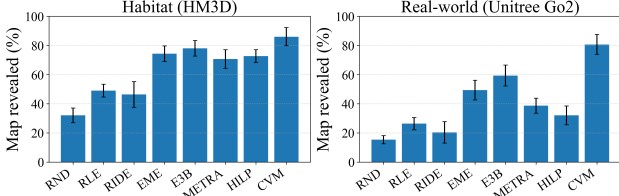

*Figure 6.* Navigation exploration coverage. Map revealed (%, mean over easy/medium/hard) on Habitat HM3D (left) and a real-world Unitree Go2 setup (right). Error bars show standard error across 5 runs. CVM achieves the highest coverage in both settings.

*Figure 7.* **Real-robot bimanual manipulation tasks with increasing difficulty.** (a) Grasp: grasp a single object using one-hand only. (b) Handover: transfer an object from the left gripper to the right gripper. (c) Targeted Handover: identify a designated target object among distractors, grasp it, and perform a left-to-right handover.

ploy the learned policy on a Unitree Go2 quadruped. The robot is equipped with a front-facing RGB-D camera and test navigation in three indoor environments of increasing difficulty (Fig. 4): **Easy** (structured room with sparse furniture), **Medium** (office-like space with moderate clutter), and **Hard** (multi-room layout with occlusions, distractors, and narrow passages). We train CVM directly on a collected real-world offline dataset, after training, we deploy the learned policy on the robot for online evaluation in the three environments.

**Embodied Manipulation Tasks.** To assess task-level exploration and control in contact-rich settings, we evaluate CVM on real-world bimanual manipulation using the ALOHA platform. We follow a fully real-data protocol: all methods are trained on the same fixed set of teleoperated demonstrations. The policy receives multi-view RGB observations and proprioception and outputs bimanual control commands. We consider three tasks of increasing difficulty (Fig. 7): **Easy** (single-object grasping), **Medium** (bimanual

handover), and **Hard** (targeted handover in clutter), where the target must be identified among distractors before grasping and transferring. Each trial randomizes object poses and clutter layouts, requiring generalization across configurations and contacts. Additional implementation details are provided in Appendix C.

### 4.2. Experimental Results and Analysis

**Exploration Coverage.** As Fig. 6 shows, we measure exploration coverage by the percentage of the environment map revealed, averaged over Easy/Medium/Hard settings. CVM achieves the highest coverage on both Habitat HM3D and the real-world Unitree Go2 setup, indicating that it explores a larger portion of the *reachable* state space under high-dimensional visual inputs and embodiment constraints. Notably, CVM's advantage over prior methods is larger in the real-world setting than in simulation, suggesting stronger robustness to realistic dynamics and visual

complexity. Prediction-error and count-based bonuses (e.g., RND, E3B) yield limited coverage improvements in these settings. Latent exploration baselines based on randomized objectives or latent displacement (e.g., RLE, RIDE, EME) improve coverage but tend to saturate earlier, consistent with myopic novelty and local cycling. Directional spanning methods (METRA, HILP) remain competitive in simulation but degrade more on the real robot, where many sampled directions are redundant or effectively unachievable under physical constraints.

**Overall Performance.** As shown in Table 2, CVM achieves the best overall task success rate (5 out of 6) across both embodied navigation and manipulation. In navigation, CVM consistently improves success as difficulty increases and maintains a large margin on the hardest setting, where several baselines degrade sharply. In manipulation, CVM is competitive on single-object grasping and delivers the strongest results on the more coordination- and perception-intensive handover tasks, with the largest gains on targeted handover under clutter. These trends are consistent with CVM's design: by prioritizing transitions that expand global latent coverage, it avoids local cycling and redundant directional exploration, yielding more robust exploration under realistic visual complexity and embodiment constraints.

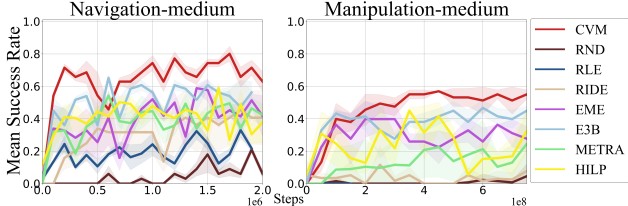

*Figure 8.* Unsupervised mean success rate comparison

**Unsupervised Performance.** To evaluate CVM in a unsupervised reward-free setting, we remove task rewards and train all methods using bonuses only, reporting task success during training. As shown in Fig. 8, CVM reaches higher success rates on both Navigation-medium and Manipulation-medium, with faster early learning and a more stable trajectory than displacement-based and directional-spanning baselines, which often plateau or fluctuate. These results indicate that CVM provides a stronger intrinsic signal that sustains exploration and translates into better task competence under realistic visual inputs and dynamics. Additional results are in Appendix B.

**Ablation Study.** Figure 5 shows that CVM's gains come from *both* the encoder and the bonus. Removing policy-mixture training (w/o PM) consistently hurts performance, especially on Nav-hard and Targeted Handover, indicating that policy-coupled encoders drift under rapidly changing exploration policies and destabilize the displacement statis-

tics used by CVM. Replacing the behavioral encoder with dynamics-based or temporal-metric variants (Dyn., Tem.) further reduces success rates, suggesting that CVM relies on a stable, behaviorally grounded latent space. Finally, replacing CVM's volume-expansion bonus with latent displacement or directional spanning (LD, DS) yields earlier plateaus and weaker results on harder tasks, consistent with these strategies failing to deliver systematic global coverage under embodiment constraints.

## 5. Related Work

Exploration is a long-standing challenge in reinforcement learning. Classical strategies include $\epsilon$-greedy (Sutton et al., 1998), count-based methods (Bellemare et al., 2016b; Ostrovski et al., 2017a; Tang et al., 2017a; Martin et al., 2017; Machado et al., 2020; Strehl & Littman, 2008; Zhao et al., 2024) and pseudo-counts based on density estimation (Bellemare et al., 2016a; Ostrovski et al., 2017b; Tang et al., 2017b; Savinov et al., 2018), and curiosity or prediction-error bonuses that reward model surprise (Pathak et al., 2017; Burda et al., 2018; Houthooft et al., 2016; Badia et al., 2020b; Bai et al., 2021; Badia et al., 2020a; Pathak et al., 2019; Flet-Berliac et al., 2021). Several model-based approaches further use learned dynamics for planning or imagined rollouts to drive exploration (Shyam et al., 2019; Ratzlaff et al., 2020; Hafner et al., 2019).

Most related to our work are *representation learning* (Yarats et al., 2021) and *latent exploration* methods, which define exploration bonuses in a learned latent space. A common design is to reward *latent displacement*, i.e., the magnitude or novelty of latent state difference, encouraging local changes in the learned representation space (Raileanu & Rocktäschel, 2020; Wang et al., 2023; Henaff et al., 2022; Jiang et al., 2025; Chiappa et al., 2023; Wan et al., 2023; Sukhija et al., 2024). Another prominent family introduces latent variables or directions to parameterize exploratory objectives and trains latent-conditioned policies; at execution time, exploration is induced either by sampling *random latent goals* (Mahankali et al., 2024; Seo et al., 2021; Eysenbach et al., 2018) or by *directional spanning*, where a direction $z$ is sampled and the agent is rewarded for progress along $z$ in latent space (Park et al., 2023; 2024; Wang et al., 2025). These approaches aim to induce diverse skills or long-horizon behaviors, but their exploration signal is tied to either direction-agnostic local novelty or externally sampled directions (spanning-based), which can be brittle in realistic embodied settings.

## 6. Conclusion

In this work, we introduced Covariance Volume Maximization (CVM), a global coverage-driven latent exploration

framework for high-dimensional and embodied reinforcement learning. CVM combines a policy-mixture behavioral encoder that reduces representation drift with an exploration bonus that rewards the exact log-determinant gain of the covariance of recent latent displacements, explicitly expanding coverage of the reachable latent region. Our theoretical analysis shows that maximizing cumulative CVM reward corresponds to maximizing a log-determinant objective, recovering the classical *D-optimal* criterion from optimal experimental design. Experiments on embodied navigation and manipulation tasks demonstrate improved exploration efficiency and robustness comparing to baseline methods.

## Acknowledgements

The work was supported by National Science and Technology Major Project (2023ZD0121401) and the Science and Technology Development Fund Macau SAR (0003/2023/RIC, 0052/2023/RIA1, 0011/2025/RIC, 001/2024/SKL for SKL-IOTSC). This work was performed in part at SICC which is supported by SKL-IOTSC, University of Macau.

## Impact Statement

This paper presents work whose goal is to advance the field of machine learning, by improving exploration in deep reinforcement learning for realistic environments with high-dimensional observations. There are many potential societal consequences of our work, none of which we feel must be specifically highlighted here.

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

# A. Proof

**Proposition 1** (On-policy objectives induce representation drift). *Let $\{\pi_t\}_{t \geq 0}$ be the evolving exploration policy. Suppose the encoder $\phi_\omega$ is trained to minimize an on-policy behavioral-metric loss*

$$\mathcal{L}_t(\omega) = \mathbb{E}_{(s,\tilde{s}) \sim \mu_{\pi_t}} \left[ \ell\big(d_\omega(s, \tilde{s}), \, d^{\pi_t}(s, \tilde{s})\big) \right]$$

*where $\mu_{\pi_t}$ is the state-pair sampling distribution induced by $\pi_t$, $d_\omega(s, \tilde{s}) = \|\phi_\omega(s) - \phi_\omega(\tilde{s})\|_2$, and $d^{\pi_t}$ denotes a policy-conditioned behavioral metric. If $\pi_t \neq \pi_{t-1}$ and either $d^{\pi_t}$ or $\mu_{\pi_t}$ depends on $\pi_t$, then the minimizer set $\arg \min_\omega \mathcal{L}_t(\omega)$ generally differs from $\arg \min_\omega \mathcal{L}_{t-1}(\omega)$. Consequently, the induced latent geometry can drift over time even when the environment dynamics are fixed.*

*Proof.* We give a constructive argument showing that once either the sampling distribution $\mu_{\pi_t}$ or the target metric $d^{\pi_t}$ changes with the policy, the encoder objective becomes a *moving target*, and its set of minimizers will typically change.

Assume the loss $\ell(\cdot, \cdot)$ is minimized at equality, i.e., for any fixed $y$, the set $\arg \min_x \ell(x, y)$ contains $x = y$ (this holds for standard choices such as squared loss $\ell(x, y) = (x - y)^2$ or Huber loss). Consider first the case where the target metric changes with the policy, i.e., $d^{\pi_t} \neq d^{\pi_{t-1}}$ on a set of state pairs with nonzero measure under $\mu_{\pi_t}$.

**Case 1: $d^{\pi_t}$ changes with $\pi_t$.** Let the state space contain at least two states $s_1, s_2$ and consider training on the single pair $(s_1, s_2)$ (this corresponds to restricting $\mu_{\pi_t}$ to put all mass on that pair, which is possible in an idealized argument and suffices to establish the claim). Let the encoder family be $\phi_\omega(s_1) = 0$ and $\phi_\omega(s_2) = \omega \in \mathbb{R}$, so that

$$d_\omega(s_1, s_2) = \|\phi_\omega(s_1) - \phi_\omega(s_2)\|_2 = |\omega|.$$

With squared loss, the on-policy objective becomes

$$\mathcal{L}_t(\omega) \;=\; \big(|\omega| - d^{\pi_t}(s_1, s_2)\big)^2.$$

Hence the minimizer set is

$$\arg \min_\omega \mathcal{L}_t(\omega) \;=\; \{\pm d^{\pi_t}(s_1, s_2)\}.$$

If $\pi_t \neq \pi_{t-1}$ and $d^{\pi_t}(s_1, s_2) \neq d^{\pi_{t-1}}(s_1, s_2)$, then these minimizer sets are different:

$$\{\pm d^{\pi_t}(s_1, s_2)\} \;\neq\; \{\pm d^{\pi_{t-1}}(s_1, s_2)\}.$$

Therefore, the optimal encoder parameters shift as the policy changes, even though the environment dynamics are fixed. This implies the induced latent geometry (and thus latent displacements $\Delta \phi$) can drift over time.

**Case 2: $\mu_{\pi_t}$ changes with $\pi_t$.** Even if the target metric were held fixed, changing the on-policy sampling distribution generally changes the optimizer. To see this, consider two state pairs $(s_1, s_2)$ and $(s_3, s_4)$ and a one-dimensional encoder family parameterized so that $d_\omega(s_1, s_2) = |\omega|$ and $d_\omega(s_3, s_4) = |1 - \omega|$ (e.g., by setting $\phi_\omega(s_1) = 0, \phi_\omega(s_2) = \omega, \phi_\omega(s_3) = 0, \phi_\omega(s_4) = 1 - \omega$). Let the target distances be fixed constants $a, b > 0$ for these pairs. Under squared loss, the objective is a weighted least-squares form:

$$\mathcal{L}_t(\omega) = p_t\big(|\omega| - a\big)^2 + (1 - p_t)\big(|1 - \omega| - b\big)^2,$$

where $p_t \in [0, 1]$ is the mass that $\mu_{\pi_t}$ assigns to the first pair. If $\pi_t \neq \pi_{t-1}$ induces a different mixture weight $p_t \neq p_{t-1}$, then the minimizer of $\mathcal{L}_t$ generally changes (indeed, for strictly convex regions of the objective the minimizer is unique and varies continuously with $p_t$). Thus, $\arg \min_\omega \mathcal{L}_t(\omega) \neq \arg \min_\omega \mathcal{L}_{t-1}(\omega)$ for generic choices of $(a, b)$ and $(p_t, p_{t-1})$.

Combining the two cases, if $\pi_t \neq \pi_{t-1}$ and either $d^{\pi_t}$ or $\mu_{\pi_t}$ depends on $\pi_t$, then the objective $\mathcal{L}_t$ changes with $t$ and its minimizer set generally changes as well. Equality of minimizers across time can occur only in degenerate situations (e.g., when the induced targets and sampling distributions happen to make $\mathcal{L}_t \equiv \mathcal{L}_{t-1}$ or share the same minimizer set). Hence, the latent geometry induced by on-policy behavioral-metric training can drift over time even under fixed environment dynamics. $\square$

**Corollary 2** (Representation drift corrupts latent displacements). *Define the latent displacement at time $t$ using the current encoder as $\Delta\phi_t \triangleq \phi_{\omega_t}(s_{t+1}) - \phi_{\omega_t}(s_t)$. If the encoder changes from $\omega_{t-1}$ to $\omega_t$, then for the same transition $(s_t, s_{t+1})$,*

$$\Delta\phi_t = \underbrace{\phi_{\omega_{t-1}}(s_{t+1}) - \phi_{\omega_{t-1}}(s_t)}_{\text{transition-induced displacement}}$$
$$+ \underbrace{\left(\phi_{\omega_t}(s_{t+1}) - \phi_{\omega_{t-1}}(s_{t+1})\right) - \left(\phi_{\omega_t}(s_t) - \phi_{\omega_{t-1}}(s_t)\right)}_{\text{representation drift}} \tag{18}$$

*The drift term can be non-negligible under rapidly changing exploration policies, causing $\Delta\phi_t$ to reflect representation updates rather than genuine behavioral change.*

*Proof.* Fix a transition $(s_t, s_{t+1})$. By definition,

$$\Delta\phi_t = \phi_{\omega_t}(s_{t+1}) - \phi_{\omega_t}(s_t).$$

Add and subtract the previous-encoder embeddings $\phi_{\omega_{t-1}}(s_{t+1})$ and $\phi_{\omega_{t-1}}(s_t)$:

$$\Delta\phi_t = \left(\phi_{\omega_t}(s_{t+1}) - \phi_{\omega_{t-1}}(s_{t+1})\right) + \phi_{\omega_{t-1}}(s_{t+1}) - \phi_{\omega_t}(s_t)$$
$$= \left(\phi_{\omega_t}(s_{t+1}) - \phi_{\omega_{t-1}}(s_{t+1})\right) + \phi_{\omega_{t-1}}(s_{t+1}) - \phi_{\omega_{t-1}}(s_t) - \left(\phi_{\omega_t}(s_t) - \phi_{\omega_{t-1}}(s_t)\right)$$
$$= \underbrace{\left(\phi_{\omega_{t-1}}(s_{t+1}) - \phi_{\omega_{t-1}}(s_t)\right)}_{\text{transition-induced displacement}} + \underbrace{\left(\phi_{\omega_t}(s_{t+1}) - \phi_{\omega_{t-1}}(s_{t+1})\right) - \left(\phi_{\omega_t}(s_t) - \phi_{\omega_{t-1}}(s_t)\right)}_{\text{representation drift}}.$$

This yields the stated decomposition exactly.

To see why the drift term can be non-negligible, define the pointwise encoder update $\delta_t(s) \triangleq \phi_{\omega_t}(s) - \phi_{\omega_{t-1}}(s)$. Then the drift term equals

$$\delta_t(s_{t+1}) - \delta_t(s_t),$$

so its norm is bounded by

$$\left\|\delta_t(s_{t+1}) - \delta_t(s_t)\right\|_2 \leq \|\delta_t(s_{t+1})\|_2 + \|\delta_t(s_t)\|_2.$$

When the policy changes rapidly, the on-policy representation objective (and its minimizers) can shift across iterations (Proposition 1), inducing encoder updates with nontrivial $\delta_t(\cdot)$ on the visited states. In that case, $\|\delta_t(s_{t+1}) - \delta_t(s_t)\|_2$ can be comparable to, or larger than, the transition-induced term, implying that $\Delta\phi_t$ may reflect representation updates rather than the underlying transition effect. This completes the proof. $\square$

**Proposition 2** (Policy mixtures reduce objective variation). *Assume the policy-conditioned metric $d^\pi$ is Lipschitz in $\pi$ with respect to a divergence $D(\cdot, \cdot)$, i.e., $\|d^\pi - d^{\pi'}\|_\infty \leq L\,D(\pi, \pi')$ for some $L > 0$.[3] Then the mixture metric varies more smoothly than the instantaneous metric:*

$$\left\|d^{\bar\pi_t} - d^{\bar\pi_{t-1}}\right\|_\infty \leq L \sum_{k=1}^{K} w_k\, D(\pi_{t-k}, \pi_{t-1-k}). \tag{19}$$

*In particular, if the policy updates are bounded on average, the right-hand side is smaller than the variation induced by using $\pi_t$ directly, yielding a more stable target for representation learning.*

*Proof.* Recall the mixture policy (over the last $K$ snapshots) is

$$\bar\pi_t(\cdot \mid s) \triangleq \sum_{k=1}^{K} w_k\, \pi_{t-k}(\cdot \mid s), \qquad w_k \geq 0,\ \sum_{k=1}^{K} w_k = 1.$$

---

[3]This assumption is standard when $d^\pi$ is defined as the fixed point of a contraction whose expectation is taken under $\pi$.

By the assumed Lipschitz continuity of the policy-conditioned metric,

$$\left\| d^{\bar{\pi}_t} - d^{\bar{\pi}_{t-1}} \right\|_\infty \;\leq\; L\, D(\bar{\pi}_t, \bar{\pi}_{t-1}). \tag{20}$$

It remains to upper bound the divergence between consecutive mixtures. Assume $D(\cdot, \cdot)$ is *jointly convex* in its arguments (this holds for total variation distance, KL divergence, and more generally $f$-divergences). Then, by joint convexity,

$$D(\bar{\pi}_t, \bar{\pi}_{t-1}) = D\Big( \sum_{k=1}^{K} w_k\, \pi_{t-k}, \; \sum_{k=1}^{K} w_k\, \pi_{t-1-k} \Big) \;\leq\; \sum_{k=1}^{K} w_k\, D(\pi_{t-k}, \pi_{t-1-k}). \tag{21}$$

Combining (20) and (21) yields

$$\left\| d^{\bar{\pi}_t} - d^{\bar{\pi}_{t-1}} \right\|_\infty \;\leq\; L \sum_{k=1}^{K} w_k\, D(\pi_{t-k}, \pi_{t-1-k}),$$

which is exactly (19).

Finally, the right-hand side is a *weighted average* of per-step policy changes over a window of past snapshots. Thus, if policy updates are bounded on average (e.g., $D(\pi_t, \pi_{t-1})$ is small most of the time, or has bounded expectation), then the mixture target $d^{\bar{\pi}_t}$ varies more smoothly than an instantaneous target $d^{\pi_t}$, providing a more stable objective for representation learning. $\qquad\square$

**Lemma 3** (Mahalanobis interpretation). *Let $\Sigma \succ 0$ and $u \in \mathbb{R}^n$. The quadratic form $u^\top \Sigma^{-1} u$ equals the squared Mahalanobis distance of $u$ under covariance $\Sigma$ (McLachlan, 1999). Moreover, if $\Sigma = Q \Lambda Q^\top$ with $\Lambda = \mathrm{diag}(\lambda_1, \ldots, \lambda_n)$ and $\alpha \triangleq Q^\top u$, then*

$$u^\top \Sigma^{-1} u \;=\; \sum_{i=1}^{n} \frac{\alpha_i^2}{\lambda_i} \tag{22}$$

*Consequently, components along low-variance directions (small $\lambda_i$) receive larger weight, while motion along high-variance directions is down-weighted.*

*Proof.* By definition, for a positive definite matrix $\Sigma \succ 0$, the (squared) Mahalanobis distance of a vector $u \in \mathbb{R}^n$ under covariance $\Sigma$ is

$$d_M^2(u; \Sigma) \;\triangleq\; u^\top \Sigma^{-1} u,$$

which establishes the first claim.

For the eigen-decomposition, since $\Sigma \succ 0$ it admits $\Sigma = Q \Lambda Q^\top$ where $Q$ is orthogonal ($Q^\top Q = I$) and $\Lambda = \mathrm{diag}(\lambda_1, \ldots, \lambda_n)$ with $\lambda_i > 0$ for all $i$. Hence

$$\Sigma^{-1} \;=\; Q \Lambda^{-1} Q^\top, \qquad \Lambda^{-1} = \mathrm{diag}\Big( \tfrac{1}{\lambda_1}, \ldots, \tfrac{1}{\lambda_n} \Big).$$

Define $\alpha \triangleq Q^\top u$. Then

$$u^\top \Sigma^{-1} u = u^\top Q \Lambda^{-1} Q^\top u = (Q^\top u)^\top \Lambda^{-1} (Q^\top u) = \alpha^\top \Lambda^{-1} \alpha = \sum_{i=1}^{n} \frac{\alpha_i^2}{\lambda_i},$$

which proves (22).

The final statement follows immediately: each coordinate $\alpha_i$ (the component of $u$ along eigenvector $q_i$) is weighted by $1/\lambda_i$, so directions with smaller variance $\lambda_i$ contribute more to the energy, while directions with larger variance are down-weighted. $\qquad\square$

**Proposition 3** (Global objective of CVM). *Assume $\Sigma_0 \succ 0$ and the rank-one update $\Sigma_t = \Sigma_{t-1} + \Delta\phi_t \Delta\phi_t^\top$. Define the per-step CVM bonus as the log-determinant increment (11), where the closed form follows from the matrix determinant lemma (Horn & Johnson, 2012). Then the cumulative bonus telescopes:*

$$\sum_{t=1}^{T} b_t \;=\; \log\det(\Sigma_T) - \log\det(\Sigma_0). \tag{23}$$

*Consequently, maximizing cumulative CVM reward over a horizon $T$ is equivalent to maximizing the single global objective* $\log \det(\Sigma_T)$.

*Proof.* By definition of the per-step CVM bonus,

$$b_t \triangleq \log \det(\Sigma_t) - \log \det(\Sigma_{t-1}),$$

where $\Sigma_t = \Sigma_{t-1} + \Delta\phi_t \Delta\phi_t^\top$ and $\Sigma_0 \succ 0$. Since $\Sigma_{t-1} \succ 0$ and $\Delta\phi_t \Delta\phi_t^\top \succeq 0$, we have $\Sigma_t \succ 0$ for all $t$, so $\log \det(\Sigma_t)$ is well-defined.

Summing $b_t$ over $t = 1, \ldots, T$ yields

$$
\begin{aligned}
\sum_{t=1}^{T} b_t &= \sum_{t=1}^{T} \big( \log \det(\Sigma_t) - \log \det(\Sigma_{t-1}) \big) \\
&= \big( \log \det(\Sigma_1) - \log \det(\Sigma_0) \big) + \big( \log \det(\Sigma_2) - \log \det(\Sigma_1) \big) + \cdots + \big( \log \det(\Sigma_T) - \log \det(\Sigma_{T-1}) \big) \\
&= \log \det(\Sigma_T) - \log \det(\Sigma_0),
\end{aligned}
$$

where all intermediate terms cancel, establishing (23).

Finally, because $\log \det(\Sigma_0)$ is a constant independent of the policy and the collected trajectory, maximizing $\sum_{t=1}^{T} b_t$ is equivalent to maximizing $\log \det(\Sigma_T)$. □

**Lemma 4** (Log-determinant and ellipsoid volume (Boyd & Vandenberghe, 2004)). *Let* $\mathrm{Vol}(\cdot)$ *denote the $n$-dimensional Lebesgue measure (Euclidean volume) on $\mathbb{R}^n$. For any $\Sigma \succ 0$, define the covariance ellipsoid $\mathcal{E}(\Sigma) \triangleq \{x \in \mathbb{R}^n : x^\top \Sigma^{-1} x \leq 1\}$ and let $\mathbb{B}_n \triangleq \{x \in \mathbb{R}^n : \|x\|_2 \leq 1\}$ be the unit Euclidean ball, then $\mathrm{Vol}(\mathcal{E}(\Sigma)) = \mathrm{Vol}(\mathbb{B}_n)\sqrt{\det(\Sigma)}$. Consequently,*

$$\log \mathrm{Vol}(\mathcal{E}(\Sigma)) = \log \mathrm{Vol}(\mathbb{B}_n) + \tfrac{1}{2} \log \det(\Sigma) \tag{24}$$

*so $\log \det(\Sigma)$ is proportional to the log-volume of $\mathcal{E}(\Sigma)$ up to an additive constant.*

*Proof.* Let $\Sigma \succ 0$. Since $\Sigma$ is symmetric positive definite, it admits a unique symmetric square root $\Sigma^{1/2} \succ 0$ such that $\Sigma^{1/2}\Sigma^{1/2} = \Sigma$ and $(\Sigma^{1/2})^{-1} = \Sigma^{-1/2}$.

Consider the linear change of variables

$$x = \Sigma^{1/2} y.$$

Then

$$x^\top \Sigma^{-1} x = (\Sigma^{1/2}y)^\top \Sigma^{-1}(\Sigma^{1/2}y) = y^\top (\Sigma^{1/2})^\top \Sigma^{-1} \Sigma^{1/2} y = y^\top \Sigma^{1/2}\Sigma^{-1}\Sigma^{1/2} y = y^\top I y = \|y\|_2^2.$$

Therefore,

$$x \in \mathcal{E}(\Sigma) \iff x^\top \Sigma^{-1} x \leq 1 \iff \|y\|_2^2 \leq 1 \iff y \in \mathbb{B}_n,$$

i.e., $\mathcal{E}(\Sigma) = \Sigma^{1/2}\mathbb{B}_n$.

By the standard volume change-of-variables formula for linear maps (Jacobian determinant), for any measurable set $A \subset \mathbb{R}^n$ and invertible matrix $A \mapsto My$,

$$\mathrm{Vol}(MA) = |\det(M)|\,\mathrm{Vol}(A).$$

Applying this with $M = \Sigma^{1/2}$ and $A = \mathbb{B}_n$ gives

$$\mathrm{Vol}(\mathcal{E}(\Sigma)) = \mathrm{Vol}(\Sigma^{1/2}\mathbb{B}_n) = |\det(\Sigma^{1/2})|\,\mathrm{Vol}(\mathbb{B}_n).$$

Since $\Sigma^{1/2} \succ 0$, $\det(\Sigma^{1/2}) > 0$, hence $|\det(\Sigma^{1/2})| = \det(\Sigma^{1/2})$. Moreover, using $\det(\Sigma) = \det(\Sigma^{1/2}\Sigma^{1/2}) = \det(\Sigma^{1/2})^2$, we obtain

$$\det(\Sigma^{1/2}) = \sqrt{\det(\Sigma)}.$$

Combining the above yields

$$\mathrm{Vol}(\mathcal{E}(\Sigma)) = \mathrm{Vol}(\mathbb{B}_n)\sqrt{\det(\Sigma)}.$$

Taking logarithms gives

$$\log \mathrm{Vol}(\mathcal{E}(\Sigma)) = \log \mathrm{Vol}(\mathbb{B}_n) + \tfrac{1}{2} \log \det(\Sigma),$$

which is (24). Therefore $\log \det(\Sigma)$ is proportional to the log-volume of $\mathcal{E}(\Sigma)$ up to an additive constant. □

**Proposition 4** (D-optimal design). *Let $\Delta\phi_t \in \mathbb{R}^n$ denote the latent displacement at transition t, and consider an auxiliary scalar response model for an unknown parameter $\theta \in \mathbb{R}^n$:*

$$y_t = \Delta\phi_t^\top \theta + \varepsilon_t \tag{25}$$

*where $\varepsilon_t \sim \mathcal{N}(0, \sigma^2)$. Then the Fisher information matrix after $T$ transitions is*

$$\mathcal{I}_T(\theta) = \frac{1}{\sigma^2} \sum_{t=1}^{T} \Delta\phi_t \Delta\phi_t^\top \tag{26}$$

*Consequently, maximizing $\log \det(\Sigma_T)$ is exactly the classical D-optimal design criterion for the estimation problem.*

*Proof.* We derive the result for the auxiliary linear-Gaussian estimation model defined on latent displacements.

**Auxiliary linear-Gaussian model.** Let $y_t = \Delta\phi_t^\top \theta + \varepsilon_t$, $\varepsilon_t \sim \mathcal{N}(0, \sigma^2)$, where $\Delta\phi_t \in \mathbb{R}^n$ is the latent displacement at transition $t$, and $\theta \in \mathbb{R}^n$ is an unknown parameter.

Stacking the observations gives:

$$y = U\theta + \varepsilon,$$

where the $t$-th row of $U \in \mathbb{R}^{T \times n}$ is $\Delta\phi_t^\top$, and

$$\varepsilon \sim \mathcal{N}(0, \sigma^2 I_T).$$

The log-likelihood is, up to an additive constant,

$$\log p(y \mid \theta) = -\frac{1}{2\sigma^2} \|y - U\theta\|_2^2.$$

**Fisher information.** The Fisher information matrix is

$$\mathcal{I}_T(\theta) = -\mathbb{E}\left[\nabla_\theta^2 \log p(y \mid \theta)\right] = \frac{1}{\sigma^2} U^\top U = \frac{1}{\sigma^2} \sum_{t=1}^{T} \Delta\phi_t \Delta\phi_t^\top.$$

**Classical D-optimality.** In classical optimal experimental design, the D-optimal criterion chooses designs that maximize

$$\det(\mathcal{I}_T(\theta)) \quad \text{or equivalently} \quad \log \det(\mathcal{I}_T(\theta))$$

(Pukelsheim, 2006). Since

$$\mathcal{I}_T(\theta) = \frac{1}{\sigma^2} \sum_{t=1}^{T} \Delta\phi_t \Delta\phi_t^\top,$$

maximizing $\log \det(\mathcal{I}_T(\theta))$ is equivalent to maximizing

$$\log \det \left( \sum_{t=1}^{T} \Delta\phi_t \Delta\phi_t^\top \right).$$

Therefore, if

$$\Sigma_T = \sum_{t=1}^{T} \Delta\phi_t \Delta\phi_t^\top,$$

then maximizing $\log \det(\Sigma_T)$ is exactly the classical D-optimal criterion for the auxiliary estimation problem.

**Regularized / Bayesian form.**    If instead

$$\Sigma_T = \Sigma_0 + \sum_{t=1}^{T} \Delta\phi_t \Delta\phi_t^\top, \qquad \Sigma_0 \succ 0,$$

then $\Sigma_T$ is a regularized information matrix. In particular, if one places a Gaussian prior

$$\theta \sim \mathcal{N}(0, \sigma^2 \Sigma_0^{-1}),$$

then the posterior precision matrix is

$$\Sigma_0 + \sum_{t=1}^{T} \Delta\phi_t \Delta\phi_t^\top.$$

Hence maximizing $\log \det(\Sigma_T)$ corresponds to maximizing the log-determinant of the posterior precision, i.e., a Bayesian D-optimal criterion.

Therefore, the log-det objective optimized by CVM admits a classical D-optimal interpretation in the unregularized case, and a regularized/Bayesian D-optimal interpretation in the general case with fixed $\Sigma_0 \succ 0$.    □

**Corollary 3** (Closed-form marginal gain). *Let* $\Sigma(S) \triangleq \Sigma_0 + \sum_{v \in S} vv^\top$. *Then for any* $u \notin S$,

$$F(S \cup \{u\}) - F(S) = \log(1 + u^\top \Sigma(S)^{-1} u), \tag{27}$$

*matching the CVM per-step bonus in Eq.* (12).

Corollary 3 closes the loop between the set-function view and the online bonus: CVM is precisely the marginal gain of the same log-determinant objective.

**Proposition 5** (Diminishing returns of the log-determinant coverage objective). *Let* $\mathcal{U}$ *be a finite set of candidate displacements and define*

$$F(S) \triangleq \log \det\left(\Sigma_0 + \sum_{u \in S} uu^\top\right), \qquad \Sigma_0 \succ 0.$$

*Then $F$ is* monotone *and* submodular. *Equivalently, for any* $A \subseteq B \subseteq \mathcal{U}$ *and any* $u \in \mathcal{U} \setminus B$,

$$F(A \cup \{u\}) - F(A) \geq F(B \cup \{u\}) - F(B).$$

Proposition 5 states that the log-determinant objective has a simple and useful structure: *diminishing returns*. Adding a displacement provides a large gain when the explored set is still small or strongly biased toward a few directions, but yields a much smaller gain once similar directions have already been covered. This is the key reason CVM prefers to *open new directions* rather than repeatedly extend the same ones. While submodularity also implies a classical $(1 - 1/e)$ greedy approximation guarantee under a cardinality budget (Nemhauser et al., 1978), we use this result only to highlight that the CVM objective is well behaved from a coverage optimization perspective, not as a convergence guarantee for the online RL training dynamics.

# B. Additional Experimental Results

**Ablation Study.** EMA alters the strict full-history properties analyzed in theory. In practice, however, EMA is designed for a recency-weighted, numerically stable approximation that is well suited to embodied RL, and bonus under EMA remains the Mahalanobis-type form, preserving the core mechanism: favoring currently under-covered directions over the already expanded ones. Empirically, we conducted a $\beta$-ablation and evaluated an EMA-free exact accumulation variant. The results can be found in Table 3.

- Small $\beta = 0.001$, favors conservative, history-dominated coverage, slightly reducing success in harder tasks (Nav-hard 61.5%, Targeted Handover. 66.2%) while only marginally affecting easier tasks ( 1% drop)

- Large $\beta = 0.1$, favors aggressively tracking the current frontier, with minor performance drop on hard tasks (Nav-hard 68.5%, Targeted Handover 70.2%) and stable results on others

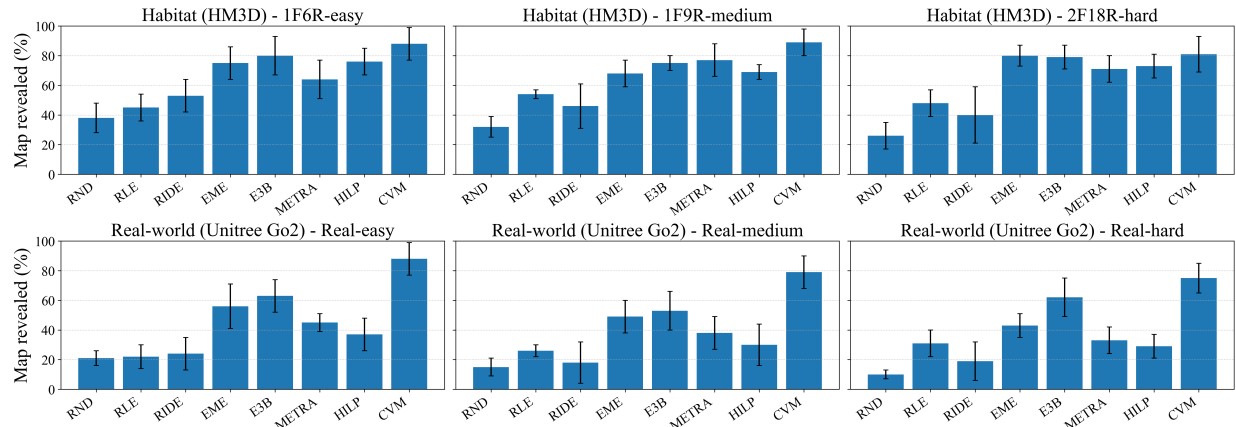

*Figure 9.* Navigation exploration coverage. Map revealed (%, mean over easy/medium/hard) on Habitat HM3D and a real-world Unitree Go2 setup. Error bars show standard error across 5 runs.

| Environment | Nav-easy | Nav-medium | Nav-hard | Grasp | Handover | Targeted Handover |
|---|---|---|---|---|---|---|
| CVM-default ($\beta = 0.01, K = 4$) | 78.7%±11.8% | 71.9%±12.1% | 69.2%±8.5% | 81.2%±7.6% | 77.8%±6.9% | 70.9%±7.5% |
| CVM ($\beta = 0.001$) | 78.1%±8.5% | 70.5%±8.9% | 61.5%±6.7% | 81.0%±5.8% | 74.5%±5.2% | 66.2%±4.5% |
| CVM ($\beta = 0.1$) | 77.9%±11.9% | 71.0%±12.5% | 68.5%±8.8% | 80.5%±7.9% | 76.9%±7.2% | 70.2%±7.7% |
| CVM (EMA-free) | 73.5%±11.3% | 65.9%±11.5% | 58.2%±7.5% | 77.5%±7.2% | 70.9%±6.5% | 60.2%±7.1% |
| CVM (K=2) | 75.0%±13.2% | 66.5%±13.5% | 63.8%±11.5% | 78.0%±8.4% | 71.3%±7.8% | 60.9%±8.0% |
| CVM (K=8) | 78.5%±11.0% | 70.1%±10.5% | 66.2%±8.0% | 81.2%±7.0% | 77.5%±5.5% | 68.8%±7.3% |
| E3B with PM | 76.5%±7.1% | 67.0%±8.8% | 40.2%±11.0% | 79.5%±7.0% | 58.8%±9.5% | 30.5%±6.0% |
| Entropy Maximization with PM | 60.5%±9.5% | 56.8%±11.2% | 41.5%±6.8% | 75.1%±8.4% | 59.4%±7.1% | 50.5%±7.0% |

*Table 3.* Mean success rates comparison (averaged over 10 seeds) in the of the embodied navigation and manipulation tasks.

- EMA-free:idealized update, but lower success on hard tasks (Nav-hard 58.2%, Targeted Handover 60.2%) due to over-reliance on early, possibly stale displacements

So performance is robust across reasonable $\beta$ values, and the EMA-free CVM variant still outperforms other methods under idealized update.

**Exploration Coverage.** Figure 9 reports detailed exploration coverage on all navigation environments, measured by the percentage of the environment map revealed. We present results separately for each difficulty level (Easy/Medium/Hard) in both Habitat HM3D and the real-world Unitree Go2 setup. Across all simulated HM3D environments, CVM consistently achieves the highest or near-highest coverage, with particularly pronounced gains in the Medium and Hard layouts. While several latent exploration baselines (e.g., EME, E3B) improve coverage over classical bonuses, their performance degrades as environment complexity increases, indicating limited scalability. Directional spanning methods (METRA, HILP) show competitive performance in Easy settings but become less reliable in harder layouts, where coverage saturates or varies significantly across runs. The advantage of CVM is more pronounced in real-world navigation. In all three physical environments, CVM substantially outperforms all baselines, achieving significantly higher coverage with lower variance. In contrast, both displacement-based and directional-spanning methods exhibit sharp performance drops, suggesting that sampling latent directions or rewarding local novelty is less effective under real-world dynamics, sensing noise, and embodiment constraints. Overall, these results reinforce the main conclusion of the paper: by explicitly rewarding global expansion of the latent displacement covariance, CVM enables more systematic and robust exploration across environment complexities, and its benefits become increasingly pronounced as tasks transition from simulation to real-world embodied settings.

**Unsupervised Performance.** Figure 10 presents unsupervised (reward-free) learning curves across all navigation and manipulation tasks. In this setting, all task rewards are removed and agents are trained purely with their respective exploration bonuses; performance is evaluated by downstream task success during training. Across all nine tasks, CVM consistently achieves the highest final success rates and exhibits faster early improvement compared to all baselines. This trend is

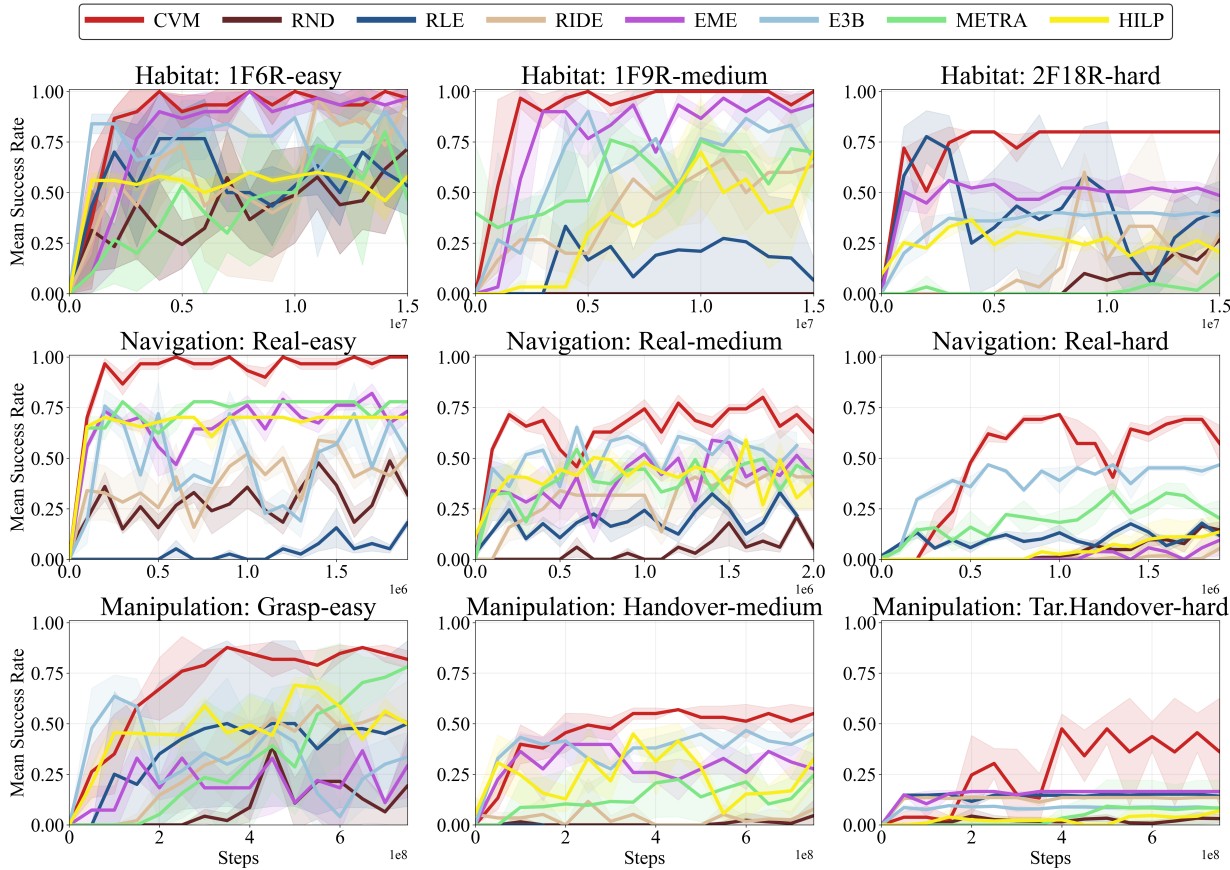

*Figure 10.* Unsupervised performance comparison across 9 tasks (averaged over 10 seeds).

particularly pronounced in medium and hard navigation environments, as well as in complex manipulation tasks such as bimanual handover and targeted handover. While several latent exploration baselines make initial progress, many plateau early or show unstable learning dynamics, especially as task complexity increases. Directional spanning methods (METRA, HILP) display competitive performance in easier settings but degrade substantially in harder navigation and manipulation tasks, where sampled directions often lead to redundant or ineffective behaviors. Displacement-based bonuses (RIDE, EME) improve local exploration but fail to sustain long-horizon progress, resulting in slower learning and lower asymptotic performance. Classical prediction-error and random-latent baselines (RND, RLE) struggle most in this reward-free regime. Overall, these results demonstrate that CVM provides a more reliable intrinsic learning signal in the absence of task rewards. By explicitly prioritizing global expansion of the reachable latent region, CVM supports sustained exploration and translates unsupervised interaction into meaningful task competence across diverse embodied settings. **Representation Drift.** We

quantitatively evaluate *representation drift* by measuring how much the latent representation of the *same* state transition changes over training as the policy evolves. Specifically, for a fixed set of transitions collected early in training, we track the average $\ell_2$ deviation between their latent displacements computed using the current encoder and those computed using the encoder at the time of data collection. Larger values indicate stronger drift, meaning that latent changes increasingly reflect encoder updates rather than true behavioral differences. Figure 11 reports the measured drift for navigation and manipulation. CVM exhibits the lowest drift in both domains, confirming that the policy-mixture behavioral objective effectively stabilizes the latent geometry under non-stationary exploration. In contrast, encoders trained with purely on-policy objectives (CVM w/o PM) show substantially larger drift, reflecting the strong coupling between representation learning and rapidly changing exploration policies. Dynamic-based and temporal metric encoders (CVM Dyn., CVM Tem.) also suffer from pronounced drift, particularly in manipulation, where visual occlusion and contact-rich dynamics amplify non-stationarity. Variants that replace the CVM objective with latent displacement or directional spanning (CVM-LD, CVM-DS) inherit this instability and exhibit consistently higher drift. Overall, these results provide quantitative evidence that stabilizing the representation is critical for reliable latent exploration. By reducing representation drift, CVM ensures

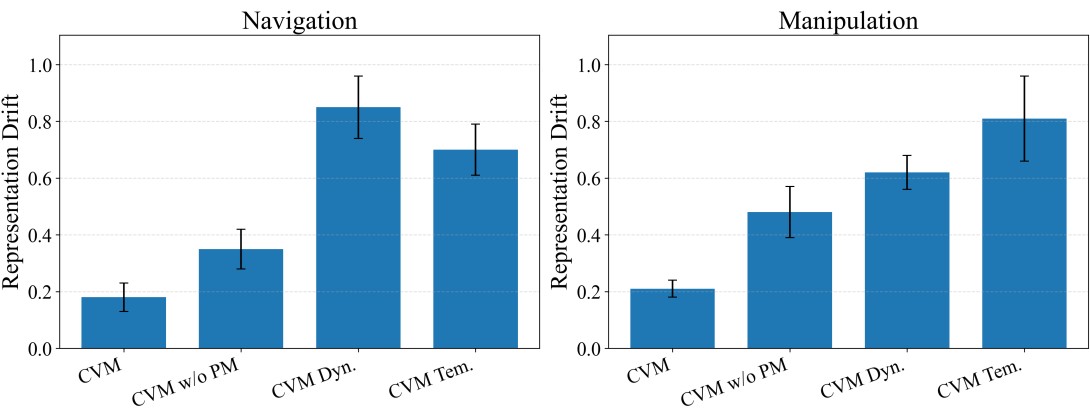

*Figure 11.* Normalized representation drift comparison between CVM and its variants (averaged over 5 seeds).

that latent displacements remain consistent and behaviorally meaningful over time, which directly supports its improved exploration coverage and downstream task performance.

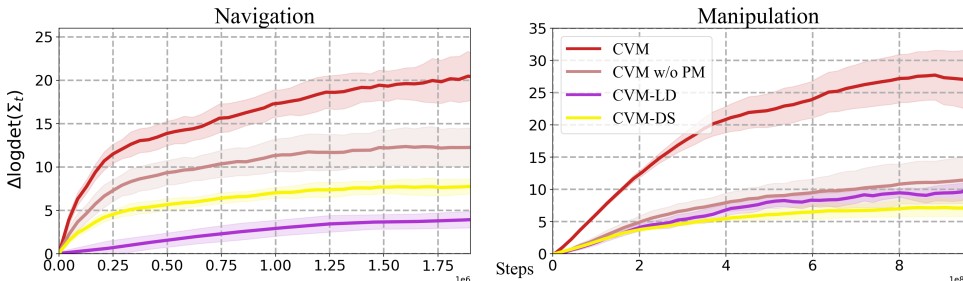

*Figure 12.* The log-det growth comparison between CVM and its variants (averaged over 10 seeds).

**Log-Determinant Growth.** We empirically verify that CVM optimizes the global coverage objective predicted by our analysis. During training, we maintain the same covariance estimator $\Sigma_t$ used by the CVM bonus (Eq. 12) and track the normalized objective $\Delta \log \det(\Sigma_t) = \log \det(\Sigma_t) - \log \det(\Sigma_0)$, $\log \det(\Sigma_t)$ computed stably via Cholesky decomposition. As the analysis shows, $\log \det(\Sigma_t)$ is proportional to the log-volume of the displacement ellipsoid, and thus measures global coverage in latent displacement space. Figure X reports $\Delta \log \det(\Sigma_t)$ on navigation and manipulation. CVM yields the fastest and most sustained log-determinant growth, indicating consistent expansion along previously under-covered directions. In contrast, removing policy-mixture training (CVM w/o PM) slows down objective growth, consistent with representation instability under rapidly changing exploration policies. Replacing CVM with local displacement bonuses (CVM-LD) or directional spanning (CVM-DS) further reduces the log-determinant growth, suggesting weaker global coverage despite inducing local novelty or direction-conditioned progress.

**Real-World Robot Evaluation Protocol.** We report real-robot navigation results on Unitree Go2 using a fixed evaluation protocol shared by all methods. Each method is evaluated on three indoor environments (Easy/Medium/Hard) described in Fig. 4. For each environment, we run $N_{\text{trial}}$ independent trials. At the beginning of every trial, the robot is placed at a predefined start region with randomized initial yaw (sampled uniformly within $[-\theta, \theta]$) and randomized goal placement within a designated goal region, ensuring comparable difficulty across methods. The policy receives the same onboard RGB-D observations used during training and outputs velocity commands at a fixed control frequency. A trial is counted as successful if the robot reaches the goal region within a time budget $T_{\max}$, i.e., the Euclidean distance to the goal is below a threshold $\epsilon$ for at least $\tau$ consecutive control steps to avoid counting transient contacts. Failures include timeouts, collisions that trigger safety stop, or leaving the valid operating area. We report mean success rate and standard error across trials for each environment, and also report the average coverage metric (map revealed percentage) computed from the onboard mapping module under the same settings for all methods.

**Compute Overhead and Wall-Clock Cost.** A common concern for log-determinant style objectives is computational

| Training Throughput and Wall-Clock Cost | | |
|---|---|---|
| Hardware | 8× RTX 4090, AMD Ryzen 9 (16 cores), 128GB RAM | |
| Metric | Environment steps/sec (higher is better) | |
| Method | steps/sec ↑ | Overhead vs. PPO ↓ |
| PPO (backbone) | 41.2k | – |
| RND (Burda et al., 2018) | 34.6k | 16.0% |
| RLE (Mahankali et al., 2024) | 39.8k | 3.4% |
| RIDE (Raileanu & Rocktäschel, 2020) | 36.9k | 10.4% |
| EME (Wang et al., 2024) | 37.5k | 9.0% |
| E3B (Henaff et al., 2022) | 38.2k | 7.3% |
| METRA (Park et al., 2023) | 37.8k | 8.3% |
| HILP (Park et al., 2024) | 37.2k | 9.7% |
| CVM (ours) | 38.6k | 6.3% |

*Table 4.* Training throughput under identical environment parallelism and network backbones. Overhead is measured relative to PPO without exploration bonus. CVM adds a small wall-clock cost due to covariance updates and Cholesky-based solves, while remaining competitive with other latent-exploration methods and substantially lighter than prediction-error/dynamics-heavy baselines.

overhead. CVM maintains a running covariance of recent latent displacements and computes the per-step bonus $b_t = \log(1 + \Delta\phi_t^\top \Sigma_{t-1}^{-1} \Delta\phi_t)$. In practice, we implement the inverse-weighted quadratic form using a Cholesky factorization of a regularized covariance $\Sigma_{t-1} + \lambda I$, solving $(\Sigma_{t-1} + \lambda I)^{-1}\Delta\phi_t$ via two triangular solves. The update is rank-one and the latent dimension is modest, so the additional compute is small relative to the backbone encoder and policy networks.

To quantify wall-clock impact, we measure training throughput in *environment steps per second* (steps/sec) under identical hardware and batch settings, and report the percentage overhead relative to the corresponding baseline implementation without CVM. Table 4 shows that CVM incurs only minor overhead while delivering substantial gains in exploration and downstream success.

## C. Implementation Details

### C.1. Environment Setting

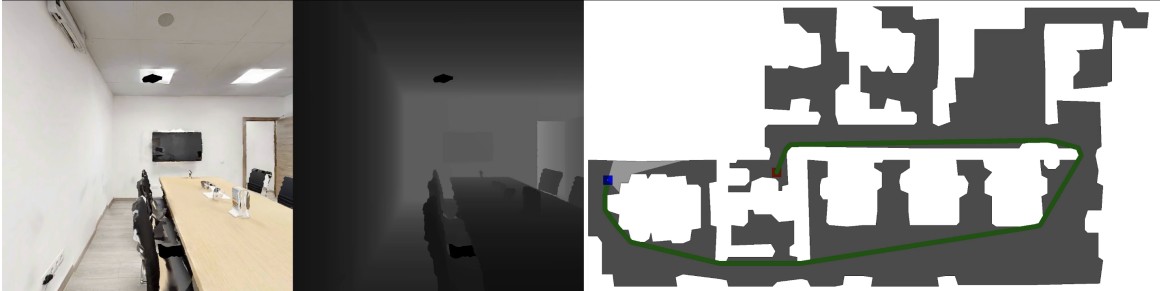

*Figure 13.* The visual observation of Habitat.

**Embodied Navigation.** We conduct navigation experiments using the Habitat platform (Savva et al., 2019), a high-fidelity simulation environment widely adopted for embodied AI research. Habitat provides a photorealistic and physically plausible 3D indoor environment, enabling agents to perceive and interact with the world via egocentric sensors such as RGB-D cameras and odometry. Agents are virtually embodied as mobile robots equipped with a forward-facing RGB-D sensor and are tasked with exploring indoor environments by issuing discrete velocity commands from a predefined action space. The 3D environments are structured via a hierarchical *SceneGraph* representation, encompassing scene geometry, objects, agent models, and sensors. The simulation backend integrates agent actions over time to update physical states and generate sensor observations at each timestep. Each episode begins with the agent randomly spawned in one of the connected rooms within a multi-room layout. The goal is to explore the environment as thoroughly as possible within a fixed number of timesteps (3000 steps per episode), without any prior knowledge of the environment layout or target location. We adopt the Habitat-Matterport 3D (HM3D) Research Dataset (Yadav et al., 2023), which contains diverse, high-resolution

reconstructions of real-world indoor environments. As illustrated in Fig. 3, we select three representative scenes with varying topological complexity. For example, the environment labeled "1F6R-easy" consists of a single floor and six rooms, representing a relatively simple layout. In contrast, "2F18R-hard" includes multiple floors and narrow corridors, presenting more significant challenges in long-horizon exploration. To evaluate spatial exploration performance, we employ the standard *coverage metric*, which quantifies the cumulative area observed by the agent's egocentric RGB-D sensors over the episode. This is computed using raycasting utilities provided by the Habitat API (Henaff et al., 2022), projecting visible areas onto a 2D floor map. The metric reflects both the diversity and spatial reach of the exploration strategy. To measure exploration coverage, we compute the area revealed by the agent's line of site using the function provided by the Habitat codebase, which uses a modified version of Bresenham's line cover algorithm. We define the exploration coverage to be:

$$\text{coverage} = \frac{\texttt{revealed area}}{\texttt{total area}} \tag{28}$$

All experiments are conducted with a fixed random seed across ten trials to ensure reproducibility.

To assess the real-world applicability and generalization capability of CVM, we deploy the trained policy on a Unitree Go2 quadruped robot and evaluate its performance in physical indoor navigation tasks. Unlike prior works that rely on simulation-to-real transfer, we adopt a fully real-world training regime: we first construct a real-world navigation dataset by teleoperating the robot through multiple indoor environments and collecting rich visual trajectories. This dataset is then used to train CVM and all baseline methods under identical conditions, enabling a fair and direct comparison of their sample efficiency and generalization ability in physical settings. We visualize the deployment of the Unitree Go2 robot in a representative multi-room indoor environment. The figure includes: (1) a first-person RGB view captured by the robot's onboard camera, showcasing perceptual diversity across scenes; (2) a third-person perspective demonstrating the robot's physical movement across the floor plan; and (3) a visualization of the robot's onboard visual-inertial SLAM and depth mapping system, highlighting the scene reconstruction capabilities from the Intel RealSense D435 RGB-D camera. The robot captures RGB images at $128 \times 128$ resolution, which are encoded by a visual encoder $\phi$ into a 32-dimensional latent embedding. This latent representation is then fed into the CVM policy, which predicts discrete SE(2) velocity commands. These are executed by the robot's native low-level locomotion controller via a wireless API interface. The entire inference pipeline runs onboard using a laptop (Intel i7 CPU + NVIDIA RTX 3060 GPU) mounted on the robot, achieving real-time control at 5 Hz. We evaluate CVM across three real-world environments of increasing complexity:

- **Easy**: An open single-room layout with clear geometric structure and minimal occlusions.

- **Medium**: An office-like indoor space with moderate visual clutter, partial occlusions, and furniture.

- **Hard**: A large multi-room environment containing narrow corridors, strong lighting changes, visual distractors, and transparent surfaces.

Each episode is initialized with randomized spawn and goal locations, requiring the policy to reason over long horizons and cope with partial observability, noisy perception, and hardware imperfections. To enable reproducible benchmarking, we curate a real-world navigation dataset consisting of 500+ successful teleoperated trajectories across all three environments, with a breakdown of 150 / 170 / 200 trajectories for Easy/Medium/Hard respectively. All methods are trained offline on this fixed dataset under identical supervision; no additional on-robot data is collected during training. Each trajectory records RGB images, proprioceptive information, and executed actions. We use this dataset to train all methods under identical supervision without any access to privileged information such as maps or global localization. This ensures that performance gains stem from algorithmic design rather than auxiliary signals. Our experiments show that the CVM-trained policy consistently reaches long-range goals with significantly higher success rates than all baselines. CVM leverages its predictive latent foresight and directional-variance-based exploration to discover globally coherent and spatially diverse paths. These findings validate that CVM not only scales to real-world deployment but also enables effective learning purely from visual data under realistic sensory and actuation conditions.

**Embodied Manipulation.** To empirically validate the robustness and dexterity of our proposed framework in physical settings, we deploy the trained policy on the ALOHA bimanual mobile manipulation system and evaluate its performance across a suite of object interaction tasks. Distinct from approaches utilizing domain randomization or simulation-to-real transfer, we adhere to a training protocol rooted entirely in real-world data. We curate a dataset of human demonstrations to facilitate a direct assessment of sample efficiency and policy precision under physical constraints. As depicted in the

experimental setup, the ALOHA workstation features two ViperX 300 leader arms and two WidowX 250 follower arms, equipped with three Logitech C922x RGB cameras positioned at the wrists, and top-down views to capture comprehensive visual context. The policy receives RGB images resized to $224 \times 224$ resolution along with a 14-dimensional proprioceptive vector representing joint positions. These inputs are encoded into a latent embedding which is then processed to predict 14-dimensional absolute joint position commands, covering the 6-DoF arm configuration and gripper state for both manipulators. The inference process is executed on a local workstation equipped with an NVIDIA RTX 4090 GPU, maintaining a control frequency of 50 Hz to ensure responsive interactions via the Dynamixel communication interface. We evaluate the policy across three task configurations of increasing complexity involving a deformable target object (a pepper):

- **Easy:** Single-Object Grasping. The workspace contains a solitary pepper on a flat table surface, requiring the robot to localize and lift the object with a designated arm under minimal visual ambiguity.

- **Medium:** Bimanual Handover. The robot must grasp the pepper with one end-effector and precisely coordinate with the opposing arm to execute a stable mid-air transfer, demanding high temporal synchronization between the dual manipulators.

- **Hard:** Cluttered Retrieval and Handover. The target object is embedded within a disorganized pile of diverse geometric distractors, necessitating selective attention to isolate the pepper from the visual clutter before executing the grasp and subsequent bimanual transfer.

Each trial begins with randomized object poses and distractor arrangements, requiring the policy to generalize over varying spatial configurations and handle contact-rich dynamics. To ensure rigorous benchmarking, we collect a real-world dataset consisting of 100-200 teleoperated trajectories for each task tier. This dataset records synchronized multi-view RGB streams, joint states, and actuator commands. We utilize this static dataset to train all comparison methods under identical supervision, prohibiting access to privileged information such as object pose tracking or depth maps. This experimental design ensures that performance differences are attributable to algorithmic architecture rather than auxiliary sensory data. Our results indicate that the policy trained with our method consistently achieves higher success rates than baseline approaches, particularly in the cluttered handover scenario. Our approach effectively leverages multi-view consistency to resolve occlusions and coordinates dual-arm motions for stable object manipulation. These findings demonstrate that our framework scales effectively to physical deployment and learns robust control strategies purely from high-dimensional visual observations.

## C.2. Computation Details

All of our experiments are conducted on 8 GPUs with 16 CPU threads, which include AMD Ryzen 9 CPU@1.20GHz (16 core) 862 CPU, NVIDIA GeForce GTX 4090Ti GPUs, and 128GB memory.

## C.3. Algorithm Pseudo code

We provide the pseudo code of CVM in Algorithm 1.

## C.4. Codebases Used

Our codebase was built atop the following codebases:

- The official RND codebase: https://github.com/openai/random-network-distillation

- The official RLE codebase: https://github.com/Improbable-AI/random-latent-exploration

- The official RIDE codebase: https://github.com/facebookresearch/impact-driven-exploration

- The official EME codebase: https://github.com/YimingWangMingle/EME

- The official E3B codebase: https://github.com/facebookresearch/e3b

- The official METRA codebase: https://github.com/seohongpark/METRA

- The official HILP codebase: https://github.com/seohongpark/HILP

---

**Algorithm 1** CVM: Covariance Volume Maximization for Latent Exploration

---

1: **Initialize:** policy $\pi_\theta$, encoder $\phi_\omega$; covariance $\Sigma \leftarrow \Sigma_0$; regularizer $\Sigma_0 = \lambda I$; EMA step $\beta$; ridge $\lambda$; mixture size $K$; mixture weights $\{w_k\}_{k=1}^K$; update interval $U$; replay buffer $\mathcal{D}$
2: **while** *not converged* **do**
3:   **for** $t = 1$ to MAX_STEP_PER_EPISODE **do**
4:     Sample action $a_t \sim \pi_\theta(\cdot \mid \phi(s_t))$ and reach $s_{t+1}$
5:     Compute latent displacement $\Delta\phi_t \leftarrow \phi_\omega(s_{t+1}) - \phi_\omega(s_t)$
6:     Compute CVM bonus: $b_t \leftarrow \log\bigl(1 + \Delta\phi_t^\top \Sigma^{-1} \Delta\phi_t\bigr)$
7:     Covariance update: $\Sigma_t = (1 - \beta)\Sigma_{t-1} + \beta\Delta\phi_t\Delta\phi_t^\top + \lambda I$
8:     Store transition $(s_t, a_t, b_t + r_t, s_{t+1})$ in rollout buffer $\mathcal{D}$
9:     **if** $t \bmod U = 0$ **then**
10:       Update policy $\pi_\theta$ via policy gradient using bonus
11:       Append current policy $\pi_\theta$ to $\mathcal{P}$ and keep last $K$ snapshots
12:       Define mixture $\bar{\pi} \triangleq \sum_{k=1}^{|\mathcal{P}|} w_k \pi_k$
13:       Sample state pairs $(s, \tilde{s}) \sim \mu_{\bar{\pi}}$ (e.g., from replay/rollouts)
14:       Update $\omega$ by minimizing $\mathcal{L}_{\mathrm{mix}}(\omega) \triangleq \mathbb{E}_{(s,\tilde{s})\sim\mu_{\bar{\pi}}}[\ell(\|\phi_\omega(s) - \phi_\omega(\tilde{s})\|_2,\ d^{\bar{\pi}}(s,\tilde{s}))]$
15:       Clear rollout buffer $\mathcal{D}$
16:     **end if**
17:   **end for**
18: **end while**

---

## C.5. RL Hyperparameters

*Table 5.* Common Hyperparameters for Habitat

| Parameter | Value |
|---|---|
| Clipping | 0.2 |
| PPO epochs | 8 |
| Number of minibatches | 32 |
| Value loss coefficient | 0.5 |
| Entropy coefficient | 5e-5 |
| Learning rate | 7e-5 |
| $\epsilon$ | $10^{-5}$ |
| Max gradient norm | 0.8 |
| Rollout steps | 128 |
| Use GAE | True |
| $\gamma$ | 0.99 |
| $\tau$ | 0.95 |
| Use linear clip decay | True |
| Maximum reward scaling | 10 |
| Hidden size | 512 |

## C.6. CVM-Specific Hyperparameters and Implementation Details

This section specifies implementation details that are critical for reproducing CVM, including the policy-mixture construction $(K, \{w_k\})$, the covariance estimator $(\beta, \lambda)$, and the encoder update schedule.

**Policy-mixture for representation learning** $(K, \{w_k\})$. CVM mitigates representation drift by training the behavioral encoder with a mixture policy $\bar{\pi}_t(a|s) = \sum_{k=1}^K w_k\pi_{t-k}(a|s)$ (Eq. (6)). In practice, we implement this mixture by maintaining a *representation buffer* that stores rollouts generated by the most recent $K$ policy snapshots.

**Snapshot cadence.** We snapshot the policy $\pi_\theta$ *once per policy update* (i.e., every $U$ environment steps). We keep the latest $K$ snapshots $\{\pi_{t-1}, \ldots, \pi_{t-K}\}$ and discard older ones.

*Table 6.* Hyper-parameters for real-world embodied navigation on Unitree Go2

| Parameters | Easy | Medium | Hard |
|---|---|---|---|
| Environment type | Single-room | Office-like | Multi-room |
| Episode horizon (steps) | 200 | 300 | 400 |
| Control frequency (Hz) | 5 | 5 | 5 |
| Action space | Discrete SE(2) velocity | Discrete SE(2) velocity | Discrete SE(2) velocity |
| Action dimension | 7 | 7 | 7 |
| Observation modality | RGB only | RGB only | RGB only |
| Image resolution | $128 \times 128$ | $128 \times 128$ | $128 \times 128$ |
| Visual encoder output dim | 32 | 32 | 32 |
| Robot platform | Unitree Go2 | Unitree Go2 | Unitree Go2 |
| Onboard sensor | Intel RealSense D435 RGB-D | Same | Same |
| SLAM / mapping | Visual-inertial (no map access) | Same | Same |
| Dataset size (teleop trajs) | $\sim$150 | $\sim$170 | $\sim$200 |
| Training regime | Offline real-world data | Offline real-world data | Offline real-world data |
| **Policy Optimization (Discrete SAC with BC regularization)** | | | |
| Policy type | Categorical over 7 actions | Same | Same |
| Replay buffer | Fixed dataset replay | Same | Same |
| Actor optimizer | Adam | Adam | Adam |
| Critic optimizer | Adam | Adam | Adam |
| Actor learning rate | $3 \times 10^{-4}$ | $3 \times 10^{-4}$ | $3 \times 10^{-4}$ |
| Critic learning rate | $3 \times 10^{-4}$ | $3 \times 10^{-4}$ | $3 \times 10^{-4}$ |
| Temperature learning rate | $3 \times 10^{-4}$ | $3 \times 10^{-4}$ | $3 \times 10^{-4}$ |
| Discount factor $\gamma$ | 0.99 | 0.99 | 0.99 |
| Target smoothing coefficient $\tau$ | 0.005 | 0.005 | 0.005 |
| Target update interval | 1 | 1 | 1 |
| Number of Q-functions | 2 | 2 | 2 |
| Minibatch size | 256 | 256 | 256 |
| Gradient updates per batch | 1 | 1 | 1 |
| Entropy tuning | Automatic | Automatic | Automatic |
| Target entropy | $0.98 \log |\mathcal{A}|$ | Same | Same |
| Initial temperature $\alpha_0$ | 0.2 | 0.2 | 0.2 |
| BC regularization weight $\lambda_{\mathrm{BC}}$ | 1.0 | 1.0 | 1.0 |
| Hidden layers (policy/Q) | [256, 256] | [256, 256] | [256, 256] |

**Mixture weights.** We use fixed weights $\{w_k\}$ throughout training. Unless otherwise stated, we use

$$w_k = \frac{1}{K} \quad \text{(uniform mixture)}$$

Uniform weights avoid introducing an additional decay hyperparameter and work well in our experiments. (Optionally, an exponential mixture can be used: $w_k \propto \rho^{k-1}$ with normalization, but we report uniform weights for reproducibility.)

**Implicit mixture via buffer sampling.** Sampling state(-transition) pairs uniformly from the representation buffer implements an implicit mixture of the data distribution induced by the recent $K$ policies. Under uniform retention of rollouts per snapshot, this matches the explicit uniform mixture above.

**Encoder update objective and state-pair construction.** The encoder $\phi_\omega$ is trained to match a mixture-conditioned behavioral metric target $d^{\bar{\pi}}$ (Eq. (7)). We optimize the encoder with a TD-style fixed-point regression consistent with Eq. (7).

**Paired transition sampling.** At each encoder update, we sample $2B$ transitions from the representation buffer: $(s_i, a_i, r_i, s'_i)$ for $i = 1, \ldots, 2B$. We then form $B$ paired transitions by a random permutation: $(s_i, a_i, r_i, s'_i)$ pairs with $(\tilde{s}_i, \tilde{a}_i, \tilde{r}_i, \tilde{s}'_i)$.

**Bootstrap target.** For each pair, we define the one-step target distance

$$y_i = |r_i - \tilde{r}_i| + \gamma \, \|\phi_\omega(s'_i) - \phi_\omega(\tilde{s}'_i)\|_2.$$

The predicted distance is $d_\omega(s_i, \tilde{s}_i) = \|\phi_\omega(s_i) - \phi_\omega(\tilde{s}_i)\|_2$.

**Loss.** We minimize a robust regression loss (Huber or squared loss):

$$L_{\mathrm{mix}}(\omega) = \mathbb{E}\big[\ell(d_\omega(s, \tilde{s}), y)\big],$$

*Table 7.* Hyper-parameters for real-world embodied manipulation on ALOHA

| Parameters | Grasp | Handover | Targeted Handover |
|---|---|---|---|
| Task difficulty | Easy | Medium | Hard |
| Episode horizon (steps) | 250 | 300 | 350 |
| Control frequency (Hz) | 50 | 50 | 50 |
| Action space | Absolute joint position | Absolute joint position | Absolute joint position |
| Action dimension | 14 | 14 | 14 |
| Observation modality | Multi-view RGB + proprio | Same | Same |
| Image resolution | $224 \times 224$ | $224 \times 224$ | $224 \times 224$ |
| Proprioceptive dimension | 14 | 14 | 14 |
| Robot platform | ALOHA (bimanual) | ALOHA (bimanual) | ALOHA (bimanual) |
| Dataset size (expert demos) | 100 | 150 | 200 |
| Training regime | Offline real-world data | Offline real-world data | Offline real-world data |
| **Policy Optimization (SAC with BC regularization)** | | | |
| Policy type | Tanh-Gaussian | Same | Same |
| Replay buffer | Fixed dataset replay | Same | Same |
| Policy optimizer | Adam | Adam | Adam |
| Q optimizer | Adam | Adam | Adam |
| Policy learning rate | $3 \times 10^{-4}$ | $3 \times 10^{-4}$ | $3 \times 10^{-4}$ |
| Q learning rate | $3 \times 10^{-4}$ | $3 \times 10^{-4}$ | $3 \times 10^{-4}$ |
| Temperature learning rate | $3 \times 10^{-4}$ | $3 \times 10^{-4}$ | $3 \times 10^{-4}$ |
| Discount factor $\gamma$ | 0.99 | 0.99 | 0.99 |
| Target smoothing coefficient $\tau$ | 0.005 | 0.005 | 0.005 |
| Target update interval | 1 | 1 | 1 |
| Number of Q-functions | 2 | 2 | 2 |
| Minibatch size | 256 | 256 | 256 |
| Gradient updates per batch | 1 | 1 | 1 |
| Entropy tuning | Automatic | Automatic | Automatic |
| Target entropy | $-|\mathcal{A}|$ | Same | Same |
| Initial temperature $\alpha_0$ | 0.2 | 0.2 | 0.2 |
| BC regularization weight $\lambda_{\mathrm{BC}}$ | 1.0 | 1.0 | 1.0 |
| Hidden layers (policy/Q) | [256, 256, 256] | Same | Same |

where we use Huber loss with $\delta = 1$ unless specified otherwise.

**Update schedule.** The encoder is updated *only at policy update time* (every $U$ environment steps), so that the encoder stays fixed during the rollout used to compute $\Delta\phi_t$. This design further reduces spurious displacement caused by within-rollout encoder drift.

**Covariance estimator $(\beta, \lambda)$ and CVM bonus computation.** Let $\Delta\phi_t = \phi_\omega(s_{t+1}) - \phi_\omega(s_t) \in \mathbb{R}^n$. We maintain an EMA covariance accumulator $\Sigma^{\mathrm{raw}} \in \mathbb{R}^{n \times n}$, initialized as $\Sigma_0^{\mathrm{raw}} = \mathbf{0}$. At each step, we compute the ridge-regularized covariance used for inversion:

$$\Sigma_t = \Sigma_t^{\mathrm{raw}} + \lambda I.$$

We then compute the CVM bonus using the previous covariance (before updating with $\Delta\phi_t$):

$$b_t = \log\left(1 + \Delta\phi_t^\top \Sigma_{t-1}^{-1} \Delta\phi_t\right).$$

After computing $b_t$, we update the accumulator by EMA:

$$\Sigma_t^{\mathrm{raw}} \leftarrow (1 - \beta)\Sigma_{t-1}^{\mathrm{raw}} + \beta\,\Delta\phi_t\Delta\phi_t^\top.$$

We compute $\Sigma^{-1}\Delta\phi_t$ via a Cholesky factorization of $\Sigma_{t-1}$ for numerical stability.

**CVM-specific hyperparameters.** Table 8 lists all CVM-specific hyperparameters used in our experiments.

# D. Use of LLM

Large language models (LLMs) were used as assistive tools for text editing and improving the clarity of exposition. They were not involved in the design of algorithms, implementation, or experimental analysis. All technical content, theoretical

*Table 8.* CVM-specific hyperparameters

| Hyperparameter | Navigation (Go2) | Manipulation (ALOHA) |
|---|---|---|
| Mixture size $K$ | 4 | 4 |
| Mixture weights $\{w_k\}$ | $w_k = 1/K$ | $w_k = 1/K$ |
| Policy snapshot cadence | every policy update | every policy update |
| Representation buffer | full offline dataset | full offline dataset |
| Buffer capacity (transitions) | $1 \times 10^5$ | $2 \times 10^4$ |
| Encoder latent dim $n$ | 32 | 32 |
| Encoder loss $\ell$ | Huber ($\delta = 1$) | Huber ($\delta = 1$) |
| Pair batch size $B$ | 256 | 256 |
| Encoder updates per policy update | 1 | 1 |
| Encoder optimizer | Adam | Adam |
| Encoder learning rate | $3 \times 10^{-4}$ | $3 \times 10^{-4}$ |
| Target network for encoder | EMA target ($\tau_\phi = 0.005$) | EMA target ($\tau_\phi = 0.005$) |
| Covariance EMA step $\beta$ | 0.01 | 0.01 |
| Covariance ridge $\lambda$ | $10^{-3}$ | $10^{-3}$ |
| Covariance initialization | $\Sigma_0^{\text{raw}} = \mathbf{0}$ | $\Sigma_0^{\text{raw}} = \mathbf{0}$ |
| Covariance inversion | Cholesky of $\Sigma + \lambda I$ | Cholesky of $\Sigma + \lambda I$ |

results, and experimental findings were produced and verified by the authors. We take full responsibility for the content of this paper. We only used LLMs for language polishing.

