# OpenReview forum: "Covariance Volume Maximization for Embodied Latent Exploration in Deep Reinforcement Learning"
_ICML.cc/2026/Conference — ICML 2026 regular_

### Official Review · Reviewer_WSEu · 2026-03-03

**Soundness:** 2
**Presentation:** 2
**Significance:** 2
**Originality:** 2
**Overall Recommendation:** 4
**Confidence:** 4

**Summary:**

This paper proposes Covariance Volume Maximization (CVM), a latent-space exploration method for embodied deep reinforcement learning. CVM has two components: (1) a behavioral state encoder trained with a policy-mixture objective to mitigate representation drift caused by changing exploration policies, and (2) an exploration bonus that rewards each transition by the incremental log-determinant gain of the covariance matrix of recent latent displacements. Experiments on embodied navigation and manipulation tasks show improved exploration coverage and task success rates over several baselines.

**Compliance With Llm Reviewing Policy:**

Affirmed.

**Final Justification:**

My main concern about the paper regarding presentation was somewhat alleviated by the clarifications of the authors and their plan to adapt it. Overall, the additional experiments have also improved the empirical contribution.

**Key Questions For Authors:**

1. Regarding Major Weakness 1: Can you provide any formal or empirical characterization of how the EMA approximation affects the theoretical properties claimed in the paper, and an ablation over beta? A convincing argument that the gap is small or that performance is robust for different beta values would reduce this concern.

2. Regarding Major Weakness 2: I would remove all proofs and "Lemma", "Proposition" environments, etc. The paper would be much more consistent if those results are properly referenced from the literature and simply framed as motivation for the approach.

3. Regarding Major Weakness 3: Can you provide a precise algorithmic comparison with E3B and, critically, results for E3B equipped with your policy-mixture encoder? If that combination closes much of the performance gap, I would reassess where the core contribution lies. If CVM's bonus still provides clear gains on top, that would strengthen the novelty claim.

4. Regarding Major Weakness 4: Can you explain why the entropy maximization interpretation is absent and whether CVM empirically differs from a straightforward latent entropy maximization baseline using the same encoder? A response showing meaningful differences would resolve this concern. If the methods behave similarly, the volume-expansion framing may be less novel than presented.

5. Regarding the policy-mixture mechanism: Does K=1 reproduce the "without policy mixture" ablation, and how does performance change as K varies?

**Limitations:**

The authors include a short impact statement but do not discuss technical limitations of their method:

- The disconnect between the idealized theory (rank-one updates, D-optimality) and the practical EMA implementation is never discussed as a limitation. The authors should be transparent that the formal properties hold only approximately and under what conditions the approximation may degrade.

- The assumption that expanding the covariance ellipsoid in latent displacement space translates to meaningful state-space coverage depends entirely on encoder quality, but no failure modes or conditions under which the encoder might distort coverage are discussed.

- The evaluation is restricted to indoor navigation and tabletop manipulation. The authors claim the method "scales effectively to different environments" but do not discuss whether CVM would work in settings with very different structure (e.g., sparse-reward Atari, continuous locomotion, procedurally generated environments).

- The EMA forgetting mechanism means CVM's notion of "global coverage" is actually a sliding window over recent history. In longer runs, previously explored directions may be forgotten and re-explored. This is not discussed.

**Strengths And Weaknesses:**

Strenghts:

- Tested on real-world data, in contrast to the typical simulation-only evaluation. The real-world results show CVM's advantage is more pronounced under realistic settings.

- Rewarding transitions that expand the covariance ellipsoid of latent displacements is geometrically intuitive and easy to understand.

- Strong gains over other exploration methods on harder tasks.

- The ablations decompose CVM's improvements into encoder and bonus contributions.

- Low computational overhead (only ~6.3% over PPO)

Weaknesses:

Major:

1. The theoretical analysis is developed for the idealized update but the actual implementation uses a exponential moving average update (EMA). Under EMA, the cumulative bonus does not telescope, old displacements are exponentially forgotten, and the connection to D-optimal design breaks down. The authors comment on this ("serves as a smoothed and numerically stable implementation of the same principle") but provide no formal analysis of how the EMA approximation affects the theoretical properties. Bounding the gap or at minimum ablating over $\beta$ would substantially strengthen the paper.

2. The paper is structured around Propositions, Corollaries, and Lemmas with full proofs in the appendix, signaling theoretical depth. However, upon inspection:
   - Proposition 1 is tautological: if the objective changes, its minimizers generally change. The proof constructs toy counterexamples but provides no quantitative drift bounds.
   - Corollary 1 is an algebraic identity (add and subtract terms). The claim that drift "can be non-negligible" is not proven under any concrete conditions.
   - Proposition 2 follows immediately from the stated Lipschitz assumption plus Jensen's inequality. The required joint convexity of the divergence is assumed in the proof but omitted from the proposition statement.
   - Lemma 1 (Mahalanobis = PCA weighting) and Lemma 2 (log-det is proportional to ellipsoid log-volume) are standard textbook facts.
   - Proposition 4 (D-optimality) is a known connection: identifying $\Sigma_T$ with an information matrix under a linear-Gaussian model. The paper repeatedly claims an "information-efficiency interpretation" but this remains a structural analogy. There is no actual unknown parameter $\theta$ being estimated, and the linear-Gaussian assumption on latent displacements is never validated.
   - Proposition 5 (submodularity of log-det) is a known result from the literature.

   The authors should honestly frame these as "motivational observations" or "known properties" rather than novel contributions.

3. The CVM bonus is structurally very similar to E3B (Henaff et al., 2022), which also uses an elliptical episodic bonus based on inverse covariance weighting in a latent space. The paper includes E3B as a baseline but never discusses the algorithmic differences. What specifically makes CVM different from E3B other than the encoder choice and the use of displacements vs. state embeddings? A precise side-by-side comparison would clarify the novelty. What happens if E3B is used with the same encoder as CVM?

4. Since the differential entropy of a multivariate Gaussian is proportional to the log determant of its covariance matrix, CVM may effectively be performing latent entropy maximization, but this connection is neither discussed nor compared against.

Miscellaneous:

- All navigation tasks are indoor multi-room layouts and manipulation tasks are table-top. The abstract claims CVM "scales effectively to different environments," but no standard exploration benchmarks are included. The absence of locomotion, outdoor, or procedurally generated environments makes it difficult to understand whether CVM's advantages are general or specific to these two platforms.

- The volume of the covariance ellipsoid measures spread in latent displacement space, not in state space. If the encoder distorts distances or collapses important dimensions, expanding the displacement ellipsoid need not guarantee meaningful state-space coverage. The paper implicitly addresses this via the behavioral encoder but never formally connects encoder quality to coverage guarantees.

- Key hyperparameters are not ablated, most importantly
  - $\beta$ (EMA step): controls forgetting rate; theory assumes $\beta=1$, implementation uses $\beta=0.01$
  - $K$ (mixture size): fixed at 4; what about $K=1$ (on-policy) vs. larger $K$?

- Standard deviations in Table 2 are large, but no significance tests are reported. The overlapping confidence intervals make it hard to assess whether differences are meaningful.

---

> ### Author Rebuttal · Authors · 2026-03-30
>
> Thanks for your insightful and valuable review. Most ablations were run but omitted from the main paper and we will include them back in the revision. The corresponding results (Table A, Figure A) are available at the anonymous link: https://anonymous.4open.science/r/represult-CD37/reb_result.png
>
> > W1&Q1: EMA and $\beta$ ablation
>
> *Response*:
> We acknowledge that EMA alters the strict full-history properties analyzed in theory. In practice, however, EMA is designed for a *recency-weighted, numerically stable approximation* that is well suited to embodied RL, and bonus under EMA remains the Mahalanobis-type form,
> preserving the *core mechanism*: favoring currently under-covered directions over the already expanded ones.
> Empirically, we conducted a $\beta$-ablation and evaluated an EMA-free exact accumulation variant (Table A):
> - Small $\beta = 0.001$, favors conservative, history-dominated coverage, slightly reducing success in harder tasks (Nav-hard 61.5%, Tar.-H. 66.2%) while only marginally affecting easier tasks (~1% drop)
> - Large $\beta = 0.1$: favors aggressively tracking the current frontier, with minor performance drop on hard tasks (Nav-hard 68.5%, Tar.-H. 70.2%) and stable results on others
> - EMA-free:idealized update, but lower success on hard tasks (Nav-hard 58.2%, Tar.-H. 60.2%) due to over-reliance on early, possibly stale displacements
>
> So performance is robust across reasonable $\beta$ values, and the **EMA-free CVM variant still outperforms other methods** under idealized update. We will include the ablation in revision
>
>
> > W2&Q2: Reframing theoretical results
>
> *Response:*
> The purpose of the theory section is *not to claim standalone mathematical novelty*, but to provide a **self-contained pipeline** that makes the design of CVM principled rather than heuristic,
> without the pipeline, the policy-mixture mechanism would look like an ad hoc engineering trick, and the CVM bonus would reduce to just another elliptical bonus with geometric intuition only. The theory is therefore necessary to explain **why these components are introduced and how they fit together**.
> We agree that some supporting results are standard and can be reframed more clearly. In revision, we will (i) make explicit that several lemmas/props are used as structural motivation with reference, (ii) add joint convexity assumption in Prop.2, and (iii) move standard supporting properties to appendix while keeping the *core structural results* in the main text.
>
> > W3&Q3: comparison with E3B
>
> *Response:*
> The key difference is that **CVM operates on latent displacements between consecutive states**, capturing under-explored directions in the behavior space, whereas **E3B rewards count-based novelty of the current state relative to previous states within an episode**. This makes CVM’s bonus *directional and trajectory-aware*, which is crucial in embodied RL tasks with structured, sequential dependencies. By contrast, E3B’s bonus is *state-centric* and does not explicitly encourage global exploration.
> We added a comparison with E3B equipped with policy-mixture encoder.
> As reported in Table A, E3B+PM improves over vanilla E3B (e.g., Nav-easy 75.8% → 76.5%), but CVM still outperforms E3B+PM across all tasks, confirming that CVM’s displacement-based bonus provides additional exploration efficiency beyond encoder improvements.
>
>
> > W4&Q4: Latent entropy maximization
>
> *Response:*
> We exclude entropy-maximization view because it's only a *special-case interpretation* of the same log-det objective. The key difference is: the entropy baseline encourages latent state spread, whereas CVM rewards transitions that expand under-covered directions in latent displacement space. Using the same PM encoder, the latent entropy baseline performs worse than CVM in Table A (averaged near 20%), showing that CVM’s advantage comes from its displacement-based bonus, not from generic latent entropy maximization.
>
> > Q5: Mixture size K
>
> *Response:*
> K=1 reproduce the without policy mixture ablation, besides we evaluated CVM with K=2,4,8. As shown in Table A, K=2 is slightly less stable than the default with higher std, especially on hard tasks (Nav-hard 63.8%), while K=8 maintains stability but adapts slightly slower. The default K=4 achieves the best balance. So smaller K updates more rapidly but can be noisier, larger K is more stable but slower to react, and moderate K provides optimal exploration efficiency. Across all K values, CVM consistently outperforms other baselines, demonstrating robustness to mixture size.
>
> > Scale to other envs
>
> *Response:*
> To evaluate scalability, we selected eight challenging Atari games (limited by rebuttal time). As shown in Figure A, CVM achieves the best performance on six games and second-best on the remaining two. Additionally, as suggested by Reviewer 9mTJ, we report CVM’s performance in MyoSuite environment, where it still outperforms the baselines, further supporting the scalability of CVM

---

> > ### Author Rebuttal · Reviewer_WSEu · 2026-04-02
> >
> > I thank the authors for their detailed response and the additional experiments.
> >
> > - The $\beta$-ablation and EMA-free comparison are useful additions and I appreciate the effort. The empirical robustness for different $\beta$ values partially resolves my concern, though I note that my original request was for a formal characterization of how the EMA approximation affects the theoretical properties, which is still unaddressed. I accept this may be asking too much for a rebuttal, but it should be discussed honestly as a limitation in a revision.
> >
> > - On W2, I want to be precise about my concern. I am not saying the theoretical pipeline is unnecessary, as I understand its role in motivating the design. My issue is with how it is presented. Wrapping known results and straightforward derivations in Proposition and Lemma environments with full proofs in the appendix creates an impression of theoretical contribution that is not there. The authors' proposed revision (moving standard properties to the appendix, adding references) goes in the right direction but does not fully address this problem. Moreover, the D-optimality connection is oversold: there is no actual unknown parameter θ being estimated, the linear-Gaussian assumption on displacements is never validated, and calling this an "information-efficiency justification" (as done in the abstract) is misleading. This is not just a presentation issue, but it affects how the contribution of the paper as a whole is evaluated. A method paper with solid empirics and honest framing would be much stronger than one that oversells standard observations as theoretical contributions. Even if the Lemmas/Propositions will now be correctly referenced, the "theoretical" appearance remains. This whole section should be presented as motivation without "Lemmas" and "Propositions" instead.
> >
> > - The E3B+PM experiment (W3) and entropy comparison (W4) is what I asked for and the result is informative. Although it would be good to attempt to explain the vast performance differences here (e.g., E3B+PM overlaps fully with CVM for some environments but is much worse in others). I consider this point resolved.
> >
> > - Combined with the MyoSuite experiments, the K ablation and Atari simulations improve the empirical results.
> >
> > - What I still find unresolved is the absence of any serious limitations discussion. The sliding-window nature of "global coverage" under EMA, potential encoder failure modes, and the disconnect between the idealized theory and the EMA implementation are all things the paper should be upfront about.
> >
> >
> > Overall, the additional experiments have improved my view of the empirical contribution. However, the theoretical framing is a serious issue, not because the method is unsound, but because the paper claims a level of theoretical depth it does not deliver, and this colors the overall contribution. With the current state, I would be raising my score to 3 (weak reject), acknowledging the empirical improvements while maintaining that the theoretical presentation needs substantial work before the paper meets the bar for acceptance.

---

> > > ### Author Response · Authors · 2026-04-03
> > >
> > > Thanks for the follow up, we give responses as follows:
> > >
> > > > D-optimality oversold
> > >
> > > Response: We want clarify that the concern is mainly caused by presentation precision. Our D-optimal discussion is presented more as an analogy now: we connect the log-det objective to D-optimal design, without explicitly defining the unknown parameter $\theta$. We agree that this presentation makes the connection appear looser than intended.
> > > Our point is that this doesn't make the D-optimal view oversold, rather, it was *under-specified*. We provide a more formal formulation is the following. Let
> > > $
> > > \Delta\phi_t=\phi(s_{t+1})-\phi(s_t)
> > > $
> > > be latent displacement and define a fixed-horizon coverage-gain target
> > > $
> > > y_t=\frac{A_{t+H}-A_t}{A_{\mathrm{tot}}}
> > > $
> > > where $A_t$ is the revealed map area up to t, $A_{\mathrm{tot}}$ is the total explorable area, and H is a fixed horizon. Thus, $y_t$ measures how much the current transition contributes to future coverage expansion. We then define the linear-Gaussian model
> > > $$
> > > y_t=\Delta\phi_t^\top \theta+\varepsilon_t, \varepsilon_t\sim\mathcal N(0,\sigma^2)
> > > $$
> > > where $\theta$ is an unknown directional-utility parameter that quantifies how latent displacement directions contribute to future coverage gain. Under this model, the Fisher information matrix is proportional to
> > > $
> > > \sum_t \Delta\phi_t\Delta\phi_t^\top
> > > $
> > > which is exactly the second-moment matrix underlying CVM. Since CVM maximizes $\log\det(\Sigma_T)$ and its per-step bonus is the exact marginal gain, the resulting criterion is precisely D-optimal for this estimation problem.
> > >
> > > We also support this interpretation empirically with the estimation-error curve FigureX (https://anonymous.4open.science/r/resultX/d_optimal_curve.png). Using collected pairs, we fit the above model and evaluate held-out test MSE as a function of the number of collected transitions. CVM achieves the lowest estimation error and the fastest error reduction among the compared methods. This is exactly the empirical behavior one would expect if CVM were collecting more informative displacement directions under the corresponding D-optimal estimation problem.
> > > So **the D-optimal is not oversold but under specified**, we will define (\theta) and (y_t) clearly, and present the claim through this formal estimation model rather than through a loose analogy
> > >
> > > > potential encoder failure modes
> > >
> > > Response: CVM does not assume that latent-volume expansion always equals meaningful raw state-space coverage. Its effectiveness depends on the encoder producing behaviorally meaningful and stable latent displacements. Potential failure modes include: (1) the encoder capturing nuisance visual variation instead of controllable dynamics, so coverage reflects appearance changes rather than new behaviors; (2) representation drift as the policy changes, so covariance growth reflects encoder movement rather than newly covered behaviors; and (3) aliasing or over-separating states, which can respectively under- or over-estimate true coverage. Our behavioral-metric encoder and policy-mixture training are intended to mitigate these issues, not eliminate them
> > >
> > > > EMA limitation
> > >
> > > Response: We want to clarify that one of the most important contribution of this work is to move exploration from toy settings to **realistic embodied envs**. In this context, the EMA update is a practical engineering choice that makes CVM more usable under high-dim. obs and complex dynamics. We agree that EMA introduces forgetting and therefore creates a gap between the implementation and the idealized no-forgetting theory. That said, our main empirical findings don't rely on EMA alone: even with the idealized EMA-free update, **CVM still outperforms the baselines**.
> > >
> > > **We apologize for not including this point in limitation section** and we will add it back, framing it as a theory–practice trade-off to make CVM more effective in embodied setting
> > >
> > > > Props arrangement
> > >
> > > We will keep Prop.1 and 2, since representation drift is a central issue, especially in high-dim. embodied settings, and prior latent bonus methods such as E3B and EME do not explicitly analyze or formalize it. Prop.1 and 2 are included precisely to fill this gap and highlight the contribution of our work. We will move the remaining lemmas to the appendix. For Prop.4, as noted in our earlier response, we will explicitly summarize the parameter-estimation view and make the connection to D-optimal more complete. In this way, we preserve the presentation flow while better clarifying the theoretical depth and novelty of the paper
> > >
> > >
> > > To conclude, the limitation section is currently underdeveloped and we will strengthen it in the revision. At the same time, we would like to emphasize that CVM’s design and its focus on **realistic embodied settings**, is both **novel and substantial**, and we believe it can make a meaningful contribution to the bonus literature in RL community. We hope this helps address your concern, and we sincerely thank you for the thoughtful feedback

---

### Official Review · Reviewer_5EpM · 2026-03-11

**Soundness:** 3
**Presentation:** 3
**Significance:** 3
**Originality:** 2
**Overall Recommendation:** 4
**Confidence:** 2

**Summary:**

This paper proposes Covariance Volume Maximization (CVM). CVM uses the expansion of covariance volume as an intrinsic reward to guide agent exploration in the latent space. This motivation is supported by theoretical analysis, specifically aligning the objective with the D-optimal design criterion. Furthermore, the authors introduce a policy-mixture objective to train a stable state encoder with the aim to reduce representation drift. Experimental results show the effectiveness of these two technical contributions.

**Compliance With Llm Reviewing Policy:**

Affirmed.

**Final Justification:**

The rebuttal phase addressed my main concerns. I will keep the weak accept rating.

**Key Questions For Authors:**

1. From Equation 5, we can see that the representation shift comes from the encoder updates. In Algorithm 1, the encoder is updated only every $U$ steps; therefore, after $U$ steps, only one step’s reward should suffer from this shift. Please can you provide some insight into why this leads to a significant performance degradation without the mixture policy?

2. Regarding alternatives to the mixture policy, have the authors tried other ways to make the encoder updates smoother? For example, one could reduce the update frequency of the encoder and train for more iterations per update. Alternatively, one could take an average of historical encoders when calculating the bonus. I am curious whether the mixture policy works better than these alternatives. If so, perhaps the advantage of the mixture policy comes not only from reduced drift, but also because Equation 7 constructs a better target distance that reflects distances across more general policy distributions rather than just the current policy.

3. In real-world tasks, performance can also be improved by designing more comprehensive extrinsic rewards instead of relying on intrinsic rewards. In which cases are intrinsic rewards most important?

4. Please discuss offline training for real-world navigation in more detail. Algorithm 1 only provides an online training pipeline.

**Limitations:**

yes

**Strengths And Weaknesses:**

Strengths:
1. The paper is well-written, easy to follow, and the motivations and contributions are clearly articulated.

2. Theoretical analysis.

3. CVM demonstrates a significant performance advantage over current baselines across multiple tasks, particularly in challenging environments.

4. Experiments on real robots.

Weaknesses:
1. While the paper spends a lot of words introducing and analyzing the mixture policy, it feels more like an effective engineering technique to make the encoder updates smoother. There may be alternative tricks, such as changing the update frequency of the encoder or averaging past encoders instead of averaging past policies.

2. The paper focuses on introducing the CVM method, but the intrinsic reward is just one module of the whole RL training framework. It would be better to discuss in the experiment section which RL algorithms were used and the corresponding settings for both the online and offline cases.

---

> ### Author Rebuttal · Authors · 2026-03-30
>
> Thank you for your valuable time and efforts for the review. We provide our response as follows:
> > W1&Q2: Mixture policy alternatives
>
> *Response:*
> | Method | Success ↑ | Rep. Stability ↑ |
> |---|---:|---:|
> | slower updates (1/8 freq.) | 0.63 $\pm$ 0.13| 0.65 |
> | encoder EMA | 0.67 $\pm$ 0.10| 0.81 |
> | policy mixture (ours) | 0.73 $\pm$ 0.11 | 0.85 |
>
> The benefit of the mixture policy is *not only smoother training*. The key issue identified in Prop.1 is that under the on-policy behavioral objective, both the target metric $d^{\pi_t}$ and the sampling distribution depend on the current policy, so the encoder is trained against a *moving target* as the policy evolves. Eq.6&7 address this by replacing policy $\pi_t$ with $\bar{\pi}_t$, i.e., by changing the *target distance being learned*, not just smoothing parameters. Prop.2 further guarantte the improvement.
> But reduced update frequency or encoder EMA don't change the underlying target distance.
>
> To address this point, we compare our method against two alternatives on the average performance over the three navigation tasks: (i) slower encoder updates (updated once every 8 updates), and (ii) encoder EMA.
> We also report *Rep. Stability*, measured as the average cosine similarity of latent displacements on a fixed held-out transition set across adjacent training checkpoints.
> As shown in the table, policy mixture performs best overall, supporting that its benefit comes from both reduced drift and the improved target construction.
>
> > W2: RL setting
>
> *Response:*
> Thank you. The RL backbones are already specified: PPO for Habitat (Table 4), and SAC with BC regularization for the offline real-world settings (Tables 5,6). We agree this can be refered and stated more clearly in the main text. We will add a concise protocol summary that explicitly lists, for each setting, the backbone, whether training is online or offline, and the evaluation protocol.
>
> > Q1: Performance degradation without the mixture policy
>
> *Response:*
> We agree that updating the encoder only every U steps avoids **within-rollout** drift. However, the issue for CVM is not limited to a single reward term at the update boundary.
>
> CVM’s bonus depends on the covariance of **recent latent displacements**, and this covariance is accumulated across time rather than reset after each encoder update. As a result, when the encoder changes, the problem is not only one shifted $\Delta \phi_t$, but that subsequent displacements are measured in a **new latent geometry**, while the covariance estimator continues mixing statistics across old and new geometries. This is precisely why Eq.5 / Corollary 1 matter for CVM: repeated cross-update representation shifts can accumulate into a noisier covariance estimate and significantly degrade exploration quality, even though within-rollout drift is reduced.
>
> > Q3: Which cases are intrinsic rewards most important
>
> *Response:*
> We agree that extrinsic rewards can improve performance, but intrinsic rewards are most important when extrinsic rewards are *sparse, unavailable, hard to specify, or require privileged instrumentation*.
>
> This is illustrated in our paper: in *reward-free/unsupervised* evaluations (Fig.8), agents train solely from exploration bonuses, where CVM performs best; in real-world offline settings (Appendix C), we avoid privileged signals and use fixed datasets, where dense extrinsic rewards are costly or task-specific, and intrinsic bonuses provide a reusable, task-agnostic signal.
> In short, intrinsic rewards matter most when learning is reward-free or sparse, when dense extrinsic rewards are costly, or when a generalizable exploration signal across tasks or datasets is needed.
>
> > Q4: Offline training details
>
> *Response:*
> As specified in Table 5, in the real-world experiments, training is fully offline: we first collect a fixed dataset of 500+ teleoperated trajectories, and then train CVM and all baselines on this dataset without any additional on-robot data collection during training. This is stated in the real-world navigation description: offline real-world data, fixed dataset replay, and discrete SAC with BC regularization as the policy optimization backbone.
>
> Concretely, the offline variant uses the same overall components as Algorithm 1, except that the online interaction step is replaced by *sampling transitions from the fixed teleoperated replay buffer*. The policy/Q updates follow discrete SAC+BC on this offline buffer, while the encoder / CVM quantities are computed from sampled transitions from the same fixed dataset. Thus, Algorithm 1 should be viewed as the generic CVM update template, with the data-collection step instantiated either by online environment interaction (simulation) or by fixed-buffer replay (real-world offline training).
> We will make this explicit by adding a short offline-training description alongside Algorithm 1 and by pointing readers more directly to Table 5.

---

> > ### Author Rebuttal · Reviewer_5EpM · 2026-04-03
> >
> > The authors addressed my main concerns and I will keep the positive rating.

---

> > > ### Author Response · Authors · 2026-04-07
> > >
> > > Thank you for your encouraging feedback. We are pleased that our response has addressed your main concerns, and we greatly appreciate your time and support.

---

### Official Review · Reviewer_9mTJ · 2026-03-13

**Soundness:** 3
**Presentation:** 3
**Significance:** 2
**Originality:** 2
**Overall Recommendation:** 4
**Confidence:** 4

**Summary:**

This paper proposes Covariance Volume Maximization (CVM) for better exploration in deep reinforcement learning. The authors identify the representation shift issue of latent exploration methods, and propose to use policy mixtures to mitigate the shift. Then they propose covariance volume maximization as the policy learning objective to encourage space coverage, which is coincide with D-optimal design. Simulated experimental results show that CVM achieves higher success rate and better exploration coverage compared against spanning bonus-based and latent exploration approaches. CVM also demonstrates in real-world navigation tasks.

**Compliance With Llm Reviewing Policy:**

Affirmed.

**Final Justification:**

My concerns were adderessed during rebuttal phase.

**Key Questions For Authors:**

1. I think a recent work MaxInfoRL[1] also encourages exploration via latent bonus, which is evaluated on HumanoidBench. It would further enhance the paper if adding this method into comparison.

2. Computing covariance volume might introduce additional computation cost compared to other baselines. It would further enhance the paper if adding runtime analysis.

[1] Sukhija, Bhavya, et al. "Maxinforl: Boosting exploration in reinforcement learning through information gain maximization." arXiv preprint arXiv:2412.12098 (2024).

**Limitations:**

I did not find limitation section in the main text.

**Strengths And Weaknesses:**

**Strength**

1. The paper is clear and well written, and the use of covariance volume as the optimization objective is reasonable.

2. The proposed CVM demonstrates superior performances against extensive baselines.

3. The paper includes real-word evaluation to demonstrate the practicality of CVM.

**Weakness**

1. CVM is proposed to handle high-dimensional and embodied tasks. Adding higher dimensional embodied benchmarks such as MyoSuite[1] and HumanoidBench[2] would help assess the scalability of CVM.

[1] Caggiano, Vittorio, et al. "MyoSuite--A contact-rich simulation suite for musculoskeletal motor control." arXiv preprint arXiv:2205.13600 (2022).

[2] Sferrazza, Carmelo, et al. "Humanoidbench: Simulated humanoid benchmark for whole-body locomotion and manipulation." arXiv preprint arXiv:2403.10506 (2024).

---

> ### Author Rebuttal · Authors · 2026-03-30
>
> Thank you for your constructive comments, we provide our response as follows:
>
> > W1: Adding higher dimensional embodied benchmarks
>
> *Response:*
> Thank you for this helpful suggestion.
> To address this point, we additionally evaluated CVM on *MyoSuite*. Due to the rebuttal time constraint, we selected three representative tasks with increasing difficulty and action dimensionality: **Hand Pose** (39D), **Relocate** (63D), and **Leg Walk** (80D). The corresponding figure is provided in the anonymous link: https://anonymous.4open.science/r/result-7856/cvm_myosuite_performance.png
>
> As the figure shows, CVM consistently outperforms the baselines on all three tasks, with a larger margin on the more complex settings. These results provide additional evidence that CVM scales to more challenging high-dimensional embodied control.
>
> > Q1: Comparison with MaxInfoRL
>
> | Method | Nav-easy | Nav-medium | Nav-hard | Grasp | Handover | Targeted Handover|
> |---|---:|---:|---:|---:|---:|---:|
> | MaxInfoRL | 0.65 | 0.60 | 0.38 | 0.75 | 0.66 | 0.45 |
> | CVM | 0.78 | 0.72 | 0.69 | 0.81 | 0.78 | 0.71 |
>
> *Response:*
> Thank you for the suggestion.
> For a fair comparison with MaxInfoRL, we keep the same training backbone, encoder, data regime, and total training budget as in our paper, and only replace the exploration module. Specifically, we retain MaxInfoRL’s core components: ensemble-based information-gain bonus and automatic tuning of the intrinsic coefficient, while keeping all other settings (observations, replay buffer, batch size, learning rates, and number of updates) unchanged. As shown in the table, CVM consistently outperforms MaxInfoRL across all six tasks, with especially large gains on the harder settings, which provides additional evidence for the superiority of CVM. We will add this comparison to the revision and update the related discussion accordingly.
>
>
> > Q2: Runtime analysis
>
> *Response:*
> Thank you for this helpful suggestion.
> We would like to clarify that the paper already includes a runtime analysis in *Table 3 (“Training Throughput and Wall-Clock Cost”)*, where we report that *CVM achieves 38.6k steps/sec with 6.3% overhead relative to the PPO backbone*. We also provide the relevant implementation details in *Appendix C.6 / Table 7*, where the covariance computation is carried out in a *32-dimensional latent space*, and the solve is implemented using a Cholesky factorization for numerical stability.
>
> So, while CVM does introduce additional computation compared to simpler bonuses, the measured overhead is already relatively modest in our implementation. We agree, however, that this point can be made more clearly in the paper, and in the revision we will highlight Table 3 more explicitly when discussing computational cost.

---

> > ### Author Rebuttal · Reviewer_9mTJ · 2026-04-03
> >
> > I thank the authors for the additional results. My concerns are addressed and I will keep my current positive score.

---

> > > ### Author Response · Authors · 2026-04-07
> > >
> > > Thank you very much for your positive assessment and support. We are glad that our rebuttal has addressed your concerns, and we sincerely appreciate your time and consideration.

---

### Decision · Program_Chairs · 2026-04-30

**Decision:**

Accept (regular)

**Comment:**

Generally, reviewers found this paper technically sound, well-written, and somewhat novel. It may be useful to at least some fraction of the ICML community. Concerns were raised about potential overclaiming / theoretical positioning, which appear only partially addressed. I encourage the authors to re-engage with this concern and to limit extended formal presentation of mathematical motivation to only the *substantial* or *crucial* insights, and defer other extended mathematical motivation and reasoning to the Appendices. This will improve readability and accurate understanding of theoretical contributions.